# Genetic surveillance of *Plasmodium falciparum* populations following treatment policy revisions in the Greater Mekong Subregion

Varanya Wasakul[1], Tess D. Verschuuren[1], Nguyen Thuy-Nhien [2], Ethan Booth[1], Huynh Hong Quang[3], Ngo Duc Thang[4], Keobouphaphone Chindavongsa[5], Siv Sovannaroth[6], Virasak Banouvong[5], Viengphone Sengsavath[5], Mayfong Mayxay[7,8,9,10], Nguyen Thi Kim Tuyen[2], Vo Ngoc Lam Phuong[2], Pham Duc Trung [2], Sónia Gonçalves[11], Soun Chen[6], Sonexay Phalivong[7], Saiamphone Xayvanghang[7], Supaporn Mahaphontrakoon[1], Richard D. Pearson [11], Paul N. Newton[7,9], Richard J. Maude [1,9,12], Elizabeth A. Ashley[7,9], Cristina V. Ariani[11], Victoria J. Simpson[11], Nicholas P. Day[1,9], Arjen M. Dondorp [1,9] & Olivo Miotto [1,9] ✉

Genetic surveillance of *Plasmodium falciparum* (*Pf*) can track antimalarial-resistant strains, to inform decision-making by National Malaria Control Programmes (NMCPs). The GenRe-Mekong project prospectively collected 5982 samples in the Greater Mekong Subregion (GMS) between 2017 and 2022, genotyping drug resistance markers, and barcodes that recapitulate genetic variation. Genotypes were analyzed with the grcMalaria R package, first described in this paper, to translate genetic epidemiology data into actionable visual information. Since 2020, *Pf* incidences decreased rapidly, accompanied by a decline of dihydroartemisinin-piperaquine (DHA-PPQ) resistant lineages, previously dominant in the eastern GMS. The frequency of *plasmepsin2/3* amplifications, conferring piperaquine resistance, dropped from 62% in 2017-2019 to 2% in 2022, coinciding with a switch in frontline therapy in Cambodia, Thailand, and Vietnam. While regional artemisinin resistance levels remained high, no evidence of emerging mefloquine resistance was found. Routine genetic surveillance proved valuable in monitoring rapid parasite population changes in response to public health interventions, providing actionable information for NMCPs.

*Plasmodium falciparum* (*Pf*), a causative agent of malaria, is responsible for more than 500,000 deaths per year, mainly in young African children[1]. Since the turn of the century, mortality has reduced, partly thanks to the availability of efficacious artemisinin-based combination therapies (ACTs), currently the first-line treatment in most endemic countries. However, parasites resistant to artemisinin and/or one of its ACT partner drugs have been reported in several regions, and have been circulating in the Greater Mekong Subregion (GMS) for over 15 years[2-8]. One of the tools used to counteract the spread of drug resistance is genetic surveillance[9,10], which can be used to map

molecular markers of resistance and conduct population-level analyses. The resulting information provides public health authorities with actionable knowledge that supports malaria control and elimination use cases, such as selecting antimalarial drug policies, monitoring drug resistance, and responding to outbreaks[11].

One of the most critical priorities for National Malaria Control Programmes (NMCPs) in the GMS is to monitor and respond to population-level changes in antimalarial drug resistance. The spread of multidrug-resistant KEL1/PLA1 strains in the eastern GMS since 2008, resulted to widespread failures of dihydroartemisinin-piperaquine (DHA-PPQ), which was the frontline treatment in Cambodia, Thailand and Vietnam[12,13]. As recommended by the World Health Organization (WHO), NMCPs in these countries switched to alternative first-line ACTs once DHA-PPQ efficacy fell below the recommended threshold of 90%[14,15]: Cambodia adopted a phased re-introduction of artesunate-mefloquine (AS-MQ) in 2017[16–19], while five provinces of Vietnam and two provinces of north-eastern Thailand adopted artesunate-pyronaridine (AS-PYR) in 2020[18–23]. The Lao People's Democratic Republic (also known as Lao PDR or Laos) continued the use of artemether-lumefantrine (AL), since its efficacy remained satisfactory[19,22,24].

The GenRe-Mekong project was established to conduct malaria genetic surveillance in partnership with NMCPs and local researchers in the GMS[9]. Parasitized dried blood spots (DBS) are collected from patients presenting at one of the dozens of partner public health facilities which form the sampling framework. DBS are then processed using the SpotMalaria amplicon sequencing platform to extract extensive genetic profiles known as Genetic Report Cards (GRC), which in turn are analyzed at population level to produce reports that are returned to NMCPs. SpotMalaria targets specific genotypes associated with antimalarial resistance (Supplementary Table 1)[9], and profiles each sample with a set of polymorphisms- known as a genetic barcode- that is informative of genetic structure and diversity in *Pf* populations[9,25–27]. To accelerate the integration of the genetic surveillance data into public health decision-making processes, NMCPs are supported with technical, genetic, and domain knowledge, to assist with data interpretation. An essential informatic tool used in these analyses is the grcMalaria library, developed by the GenRe-Mekong project and presented for the first time in this article. This R package turns genomic data into meaningful graphical visualizations that support NMCP activities, such as geographical maps of malaria resistance prevalence, diversity and relatedness. The tool is easy to use, requiring very few lines of coding, and returns results rapidly as image and spreadsheet files, making geospatial analyses accessible to users with non-technical backgrounds.

We presented early results from the GenRe-Mekong project in a previous publication[9]. Here, we provide an update, describing changes in antimalarial drug resistance epidemiology in the GMS, based on an analysis of 5982 *Pf* dried blood spot samples collected by GenRe-Mekong from symptomatic patients in Cambodia, Laos, and Vietnam between 2017 and 2022, a period of time during which several NMCPs changed their malaria frontline treatment policies.

## Results

Between January 2017 and December 2022, GenRe-Mekong collected and processed a total of 5982 *Pf* samples from 21 provinces in three countries (Fig. 1b and Supplementary Fig. 1). Sequencing coverage was high across the genotyped samples, with a mean of 747 reads across barcode loci and 565 reads across drug resistant loci (Supplementary Fig. 2). In Vietnam, the project processed 3340 samples in 2017–2022 (32.7% of WHO-reported *Pf* infections); in Laos, it processed 2400 samples over the same period (16.2%); while in Cambodia, collections took place between July 2020 and December 2022, producing 242 samples (21.0% of the WHO-reported infections in the country) (Supplementary Fig. 3)[28–30]. A rapid decline in national *Pf* prevalence occurred sequentially across the three neighboring countries: first in

Cambodia starting in 2018, then in Laos from 2019, and finally in Vietnam from 2020 (Fig. 1a). This decline in prevalence reflected a decrease in the number of confirmed cases rather than a decline in the number of tests (Supplementary Fig. 4). The longitudinal prevalence trend, as reported by WHO[28–30], has a strong correspondence with the number of samples collected by the project in Laos and Vietnam, indicating that routine surveillance data can serve as an indicator of *Pf* prevalence in these countries (Fig. 1a). As the number of cases declined over time, *Pf* populations became more spatially patchy. By 2022, the majority of samples were concentrated in one province in each country: Attapeu in Laos, Gia Lai in Vietnam, and Pursat in Cambodia (Supplementary Table 2); these provinces had consistently exhibited the highest incidence within their respective countries in previous years. Lack of coverage in certain provinces towards the end of our sampling period reflects underlying epidemiological trends rather than a lack of collection effort: although sampling sites across the whole region were engaged, the marked reduction in *Pf* cases led to some provinces not collecting any samples in the final period.

Given this evolution of the *Pf* epidemiological landscape, we opted to conduct longitudinal analyses by partitioning our sample set into three time periods (2017–2019, 2020–2021, and 2022), which reflect different epidemiological scenarios. Specifically, the period from 2020–2021 was a unique phase characterized by policy changes and major external events (e.g., pandemic-related movement restrictions), which differentiates it from both the earlier years and the year 2022. By separating these three periods, we sought to capture and highlight temporal trends more effectively, and assess how change factors have influenced *Pf* epidemiology.

### DHA-PPQ resistance decline after changes in first-line treatments

We tracked the spread of DHA-PPQ resistant strains, KEL1/PLA1, by mapping the frequency of two key markers: *pm23* amplifications conferring resistance to piperaquine (PPQ-R) and mutations in *kelch13* gene associated with resistance to artemisinin (ART-R) as listed by the WHO (Supplementary Table 1)[31]. Although several *crt* mutations are also associated with PPQ-R, they rarely occur in field samples unless accompanied by the *pm23* amplification; therefore *pm23* amplification is deemed sufficient for defining resistance due to its strong association with treatment failure[4,12]. In the 2017–2019 period, 62% (1957/3132) of all collected samples were predicted to be resistant to DHA-PPQ, as they carried both markers (Fig. 2a). Prior to that period, DHA-PPQ was used as frontline treatment for uncomplicated malaria in Cambodia, Thailand and Vietnam, which led to cross-border spread of the KEL1/PLA1 strain, resistant to both ACT components, across large areas of the eastern GMS[13]. This strain also spread into the southern Lao provinces of Attapeu and Champasak, where it introduced high levels of DHA-PPQ resistance, despite DHA-PPQ never being selected as a national treatment[9,32]. Starting in 2017, Cambodia gradually switched its frontline treatment from DHA-PPQ to AS-MQ[17–19,22], while Thailand and Vietnam switched from DHA-PPQ to other ACTs by 2020[18,19,22,23]. Following these changes, regional levels of DHA-PPQ predicted resistance fell steeply, from 62% (1957/3132) to 30% (204/690) in 2020–2021, and 1% (2/278) in 2022 ($Z = -23.9$, $p < 0.001$) (Fig. 2a). This was underpinned by a decline in predicted PPQ-R, dropping to 31% (228/737) in 2020–2021, and 2% (5/316) in 2022 ($Z = -24.5$, $p < 0.001$) (Fig. 3a and Supplementary Fig. 5).

Regional *Pf* prevalence shows a highly significant positive correlation with levels of predicted PPQ-R estimated from the proportions of *pm23* amplifications ($R = 0.64$, $p < 0.001$), where periods of lower prevalence were generally characterized by lower PPQ-R proportions, suggesting that switching away from the use of DHA-PPQ as the frontline ACT may have had an impact on prevalence (Supplementary Fig. 6). However, there are other factors that may have contributed to the decline in the number of cases: in particular, the COVID-19 pandemic

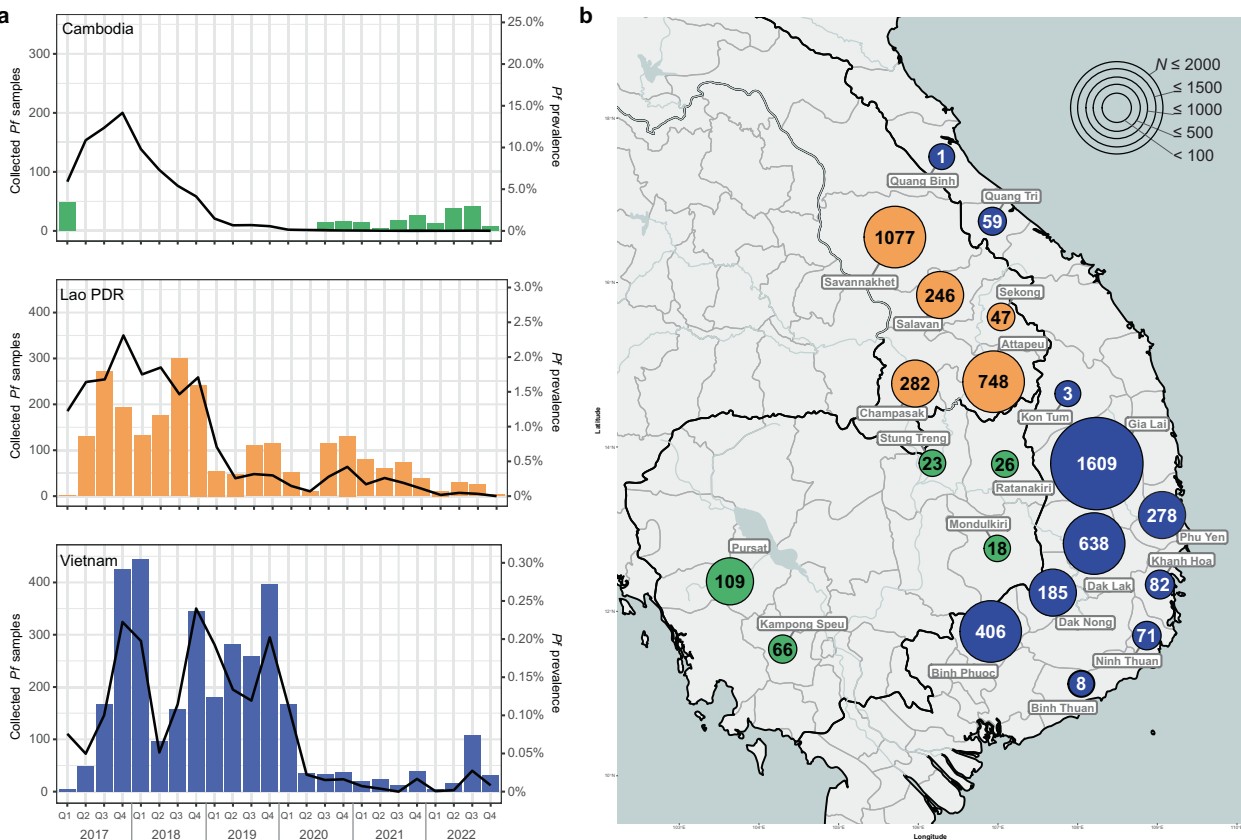

**Fig. 1 | *P. falciparum* (*Pf*) samples genotyped by GenRe-Mekong during the study period. a** Quarterly counts of *Pf* samples collected and genotyped by the GenRe-Mekong project, presented as bar charts. No routine surveillance was carried out by GenRe-Mekong in Cambodia in 2018 and 2019. Malaria prevalence, calculated as the proportion of confirmed *Pf* cases over the total number of tested cases, as reported by WHO[28–30], is shown as a black line in each chart. The prevalence scales differ between countries. **b** Spatial distributions by province of *Pf* samples collected between 2017 and 2022. Markers are colored by country, and marker size represents the number of samples from the province (*N*). Source data are provided as a Source Data file.

resulted in travel and trade restrictions that may have affected the transmission and spread of malaria parasites. To tease apart the impact of these two factors, we applied segmented regression to analyze shifts in observable trends at each intervention point; this analysis could only be conducted in Vietnam, where our data supported trend analyses before and after the two interventions. Following the change of frontline ACT, a significant reduction in prevalence was observed ($\beta = -0.1069$, $p = 0.017$), reversing the upward trend to a negative slope. Although the pandemic-related lockdown policy also shifted the trajectory from positive to negative, its immediate impact on prevalence was more modest and statistically non-significant ($\beta = -0.0526$, $p = 0.111$). These results suggest that, although both interventions impacted the decline in prevalence in Vietnam with robust explanatory power ($R^2 \approx 0.70$), the switch of frontline ACT had a more pronounced and statistically significant impact on reduction (Supplementary Fig. 7c). Data from Vietnam also revealed a strong correlation between predicted PPQ-R prevalence–measured by *pm23* amplification–and changes in first-line treatment policy. Following the replacement of DHA-PPQ with AS-PYR in five endemic Vietnamese provinces, we observed a reduction in *Pf* prevalence and parasites with *pm23* amplifications within 3 months, ultimately leading to the complete disappearance of PPQ-R parasites within 18 months (Fig. 4 and Supplementary Fig. 7c). The effect of frontline ACT changes could not be assessed in Cambodia, since their policy change took place near the beginning of our study; Laos did not change its treatment policy

during this period (Supplementary Fig. 7a, b). In both of these countries, lockdown restrictions during the pandemic showed no significant effect on prevalence, nor on PPQ-R decline (all $p > 0.05$).

In addition to *pm23* amplifications, seven *crt* mutations have been associated with reduced susceptibility to piperaquine, either in vivo or in vitro[31]. To confirm the negative trend of PPQ-R frequency in the region, we analyzed the frequencies of these *crt* mutations, to investigate whether they are present in parasites lacking the *pm23* amplification marker, and therefore predicted to be sensitive to piperaquine. One associated mutation (C350R) was completely absent from our sample set, while the remaining six mutations circulated at very low frequencies throughout the study period (Supplementary Figs. 8 and 9). No increase in the regional prevalence of these *crt* mutation was observed as *pm23* amplifications decreased in frequency. Rather, the frequencies of the two most common mutations (I218F and T93S) declined in parallel with those of the *pm23*. Moreover, *crt* mutations were significantly more likely to be detected in the presence of *pm23* amplifications ($\chi^2 = 245.2$, $p < 0.001$), suggesting these resistance markers typically co-occur rather than circulate independently (Supplementary Fig. 10). The only exception was the H97Y mutation in Cambodia, which occurred more frequently in samples without *pm23* amplification; however, the declining trend and low prevalence of H97Y (<10% frequency in 2022) are consistent with the waning of PPQ-R (Supplementary Fig. 11). The concordant evidence from multiple validated molecular markers- *pm23* amplifications and

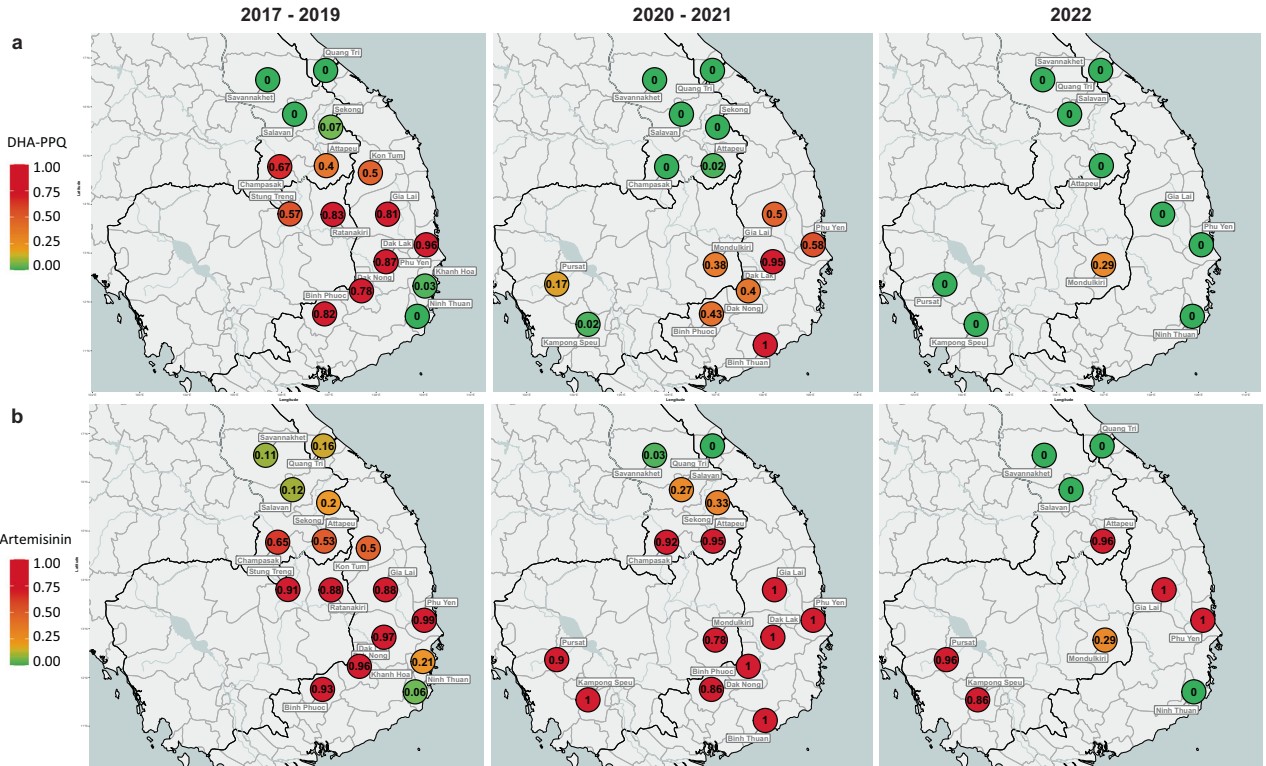

**Fig. 2 | Spatiotemporal patterns of predicted resistance to DHA-PPQ and artemisinin, from 2017 to 2022.** Predicted resistance to **a** dihydroartemisinin-piperaquine (DHA-PPQ) and **b** artemisinin at provincial level. Left: 2017–2019, middle: 2020–2021, right: January–December 2022. Resistance to artemisinin was predicted based on the presence of nonsynonymous mutations in the *kelch13* gene, while resistance to DHA-PPQ was inferred from both *kelch13* mutations and *plasmepsin2/3* gene amplification (Supplementary Table 1)[9]. Marker colors reflects resistance prevalence, ranging from 0 to 1, where 0 means no parasites were predicted to be resistant, and 1 means 100% of the parasites in the province carried the relevant resistance markers. A marker appears when at least two samples were processed from the province. Source data are provided as a Source Data file.

*crt* mutations- provides support for a regional decline in PPQ-R prevalence during this study period.

## Resistance to other antimalarials

ART-R levels remained high throughout the period analyzed (Fig. 3), except in some provinces at the periphery of the endemic region (Savannakhet in Laos, Quang Tri and Ninh Thuan in Vietnam), where most parasites remained susceptible to both artemisinin and piperaquine (Fig. 2b and Supplementary Fig. 5), suggesting that local parasite populations were isolated from the spread of DHA-PPQ resistant strains in the eastern GMS. Predicted mefloquine resistance (determined by detection of amplifications of the *mdr1* gene[33]) remained low in the region throughout the study periods (Fig. 3 and Supplementary Fig. 12), even after Cambodia adopted AS-MQ as frontline therapy in 2017[17]; in 2022, we detected no samples with an *mdr1* amplification. Markers of resistance to the historical drugs chloroquine, sulfadoxine and pyrimethamine remained high across the region throughout the study period (Fig. 3 and Supplementary Fig. 13).

## Distribution of *kelch13* allele variants across fragmented populations

It has been shown that ART-R strains in the eastern GMS originated from multiple founder populations carrying different *kelch13* alleles[34,35]. Parasite strains carrying *kelch13* haplotypes inherited from those early resistant populations have persisted over time[32], even though their geographical distribution and prevalence often changed (Fig. 5). In 2017–2019, the majority of ART-R parasites in the eastern GMS carried the *kelch13* C580Y mutation (98%, 2482/2539), and 79% of these mutants (1957/2482) were classified as KEL1/PLA1 (possessing both *kelch13* C580Y

and *pm23* amplification) (Supplementary Fig. 14). However, since 2020 the dominance of KEL1/PLA1 waned, and other *kelch13* variants, previously circulating at low frequency, expanded in the region. In the western provinces of Cambodia, *kelch13* Y493H increased in frequency, and eventually dominated the population in Pursat province in 2022 (Fig. 5 and Supplementary Table 3). In Laos, *kelch13* R539T mutants expanded in 2020–2021 in Attapeu and Champasak provinces, causing an outbreak[32], and subsequently subsided in 2022 to be replaced by parasites possessing the *kelch13* C580Y mutation without *pm23* amplification in Attapeu. In Vietnam, C580Y remained the dominant *kelch13* variant. We observed that the presence of these ART-R *kelch13* alleles was strongly associated with the NFD *mdr1* haplotype (characterized by N86, Y184F, and D1246 mutations; $\beta = 3.44$, $p < 0.001$). In contrast, wild-type *kelch13* parasites predominantly carried either the NYD (N86, Y184, D1246; wild-type) or YYD (N86Y, Y184, D1246Y) *mdr1* haplotypes ($\beta = -5.14/-6.81$, $p < 0.001$) (Supplementary Fig. 15).

## Clustering of parasite populations

Expecting the changes in *kelch13* allele distributions to be a result of the change in prevalence of circulating populations, we sought to examine population structure by clustering *Pf* barcodes by similarity and examining the geographical distributions of the clusters. Clustering identical or near-identical parasites (sharing at least 95% barcode similarity) identified 27 clusters with ≥20 members (Table 1). In the period 2017–2019, four of the five largest clusters (KLV01, KLV02, KLV03 and KLV05), accounting for 30% of samples (1398/4632), were found to carry both the *kelch13* C580Y mutation and a *pm23* gene amplification, and thus likely to have emerged from the KEL1/PLA1 strain. Two of these large clusters (KLV01 and KLV05) had considerable

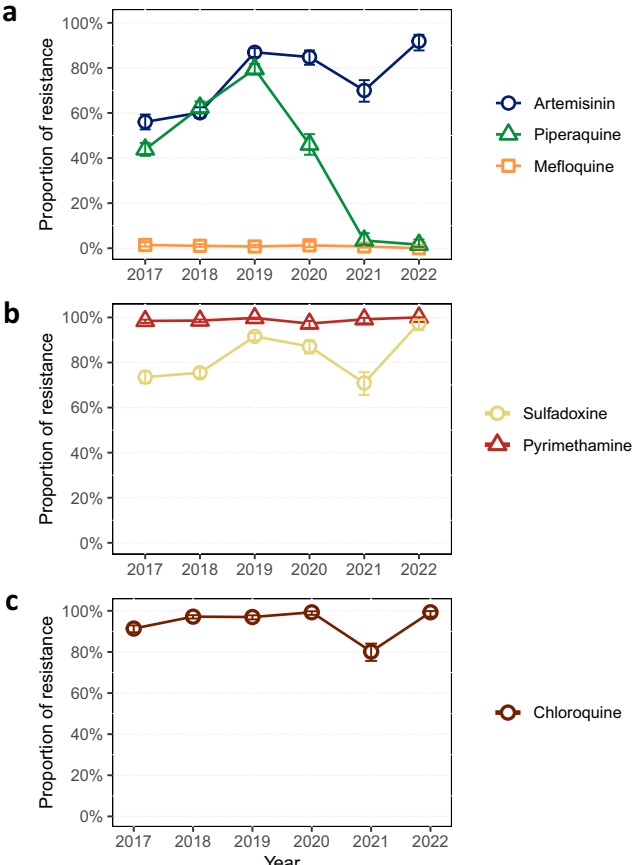

**Fig. 3 | Predicted regional antimalarial resistance from 2017 to 2022.** Panels show the trend in proportions of samples predicted to be resistant to **a** artemisinin, piperaquine and mefloquine, **b** sulfadoxine and pyrimethamine and **c** chloroquine. Resistance was predicted based on established molecular markers: nonsynonymous mutations in the *kelch13* gene for artemisinin, amplifications in *plasmepsin2/3* for piperaquine, amplifications in *mdr1* for mefloquine, the *dhps* 437G mutation for sulfadoxine, the *dhfr* 108N mutation for pyrimethamine, and the *crt* 76T mutation for chloroquine (Supplementary Table 1)[9]. Data are presented as proportion of resistant samples (resistant/total), with error bars indicating 95% confidence intervals calculated using the Wilson score interval with continuity correction. Samples with undetermined status−due to missing genotypes or mixed infections−were excluded from the analysis. For each drug, sample size (*N*) for the years 2017 to 2022 were as follows: artemisinin (*N* = 894, 1683, 1179, 521, 370, 270); piperaquine (*N* = 1177, 1289, 1207, 476, 261, 316); mefloquine (*N* = 696, 1240, 1192, 471, 377, 317); sulfadoxine (*N* = 1171, 1739, 1249, 551, 330, 239); pyrimethamine (*N* = 1225, 1724, 1237, 512, 363, 238); and chloroquine (*N* = 1124, 1728, 1255, 555, 368, 282). Source data are provided as a Source Data file.

geographic spread across the eastern GMS (Fig. 6). Following the implementation of new frontline treatment policies, however, notable changes occurred in the parasite populations. From 2020, clusters remained confined within single countries, often restricted to single provinces, and populations of non-KEL1/PLA1 clusters began to expand reaching high frequencies, as was the case for KLV04 and KLV07 in southern Laos and KLV12 in western Cambodia (Fig. 6 and Table 1). In Vietnam, the KEL1/PLA1-derived clusters KLV01 and KLV03 continued to circulate but lost their *pm23* amplification (Supplementary Table 4). By the third quarter of 2021, none of the circulating clusters possessed the *pm23* amplification, and the Central Highlands of Vietnam were dominated by the piperaquine-susceptible KLV03 population (Fig. 6). Taken together, these results suggest that the observed reduction in DHA-PPQ resistance in 2020–2021 can be explained by the disappearance of KEL1/PLA1 clusters, or their loss of *pm23* amplification, occurring after changes in first-line treatment policies.

## Discussion

In this study we analyzed the genetic epidemiology of *Pf* in the eastern GMS through longitudinal surveillance between 2017 and 2022, updating our previously published results[9]. Our data shows that the revisions of first-line treatment policies in the region were followed by important changes in the epidemiological landscape, which included: a marked decline in *Pf* infections, the near disappearance of *pm23* amplifications and PPQ-R *crt* mutations from circulating strains, the fragmentation of the artemisinin-resistant populations, and the absence of *mdr1* amplifications that would otherwise threaten AS-MQ treatment efficacy.

The prevalence of DHA-PPQ resistant parasites collapsed from 62% in 2017–2019 to 1% in 2022, the timing coinciding with the adoption of new first-line treatment strategies in three countries in the eastern GMS. In the previous decade, the prolonged use of DHA-PPQ as frontline treatment resulted in massive selective pressure on the parasites, and a rapid spread of KEL1/PLA1 lineages across the eastern GMS[12,13,36,37]. The parasite population responded just as strikingly and rapidly to the release of pressure from changes in treatment guidelines as it did to the intense DHA-PPQ pressure from previous public health policies. It is likely that, in the absence of piperaquine pressure, *pm23* amplifications and *crt* mutations associated with piperaquine resistance are costly in terms of parasite fitness, and lead to a survival disadvantage in the absence of piperaquine pressure. A recent in vitro study supported this notion by demonstrating that cultured C580Y parasites spontaneously de-amplify *pm23* when piperaquine pressure is removed[38]. Thus, the change in policy appears to have produced two different effects on KEL1/PLA1 parasites: either caused them to succumb in competition with fitter ART-R strains, which previously had been circulating at low frequency alongside KEL1/PLA1[32]; or cause them to de-amplify *pm23* while retaining their ART-R phenotype, improving their fitness under the new circumstances.

The timeline of events also strongly suggests that the introduction of efficacious replacements for DHA-PPQ, particularly in Vietnam, had a statistically significant impact, contributing to a rapid collapse in the overall number of cases[17,22–24]. Similarly, *Pf* prevalence in Cambodia started to decline rapidly in 2018 (WHO data, Fig. 1) following a change in treatment policy starting in 2017. Although our dataset does not cover Cambodian samples from 2017–2019, evidence from other studies shows that the KEL1/PLA1 population underwent sustained growth until 2018, constituting 75–100% of parasites in multiple parts of Cambodia in 2016–2018[13]. Thus, the decline in Cambodia's prevalence during 2018–2019, before any COVID-19-related movement restrictions, was not only rapid and geographically widespread but also strain-specific, as the decline of KEL1/PLA1 give way to other ART-R strains. In the southernmost provinces of Laos, the presence of KEL1/PLA1 parasites was most likely due to importation from neighboring regions of Cambodia or Thailand, where they circulated at high frequency[13,37]. Although these ART-R parasite had been successful in Laos despite DHA-PPQ not being the treatment of choice, their numbers declined as their frequencies waned in Cambodia and Thailand. It is also likely that movement restrictions due to the pandemic would have made a major contribution in reducing parasite importation, further accelerating the demise of KEL1/PLA1 in Laos and facilitating the emergence of other ART-R strains[32].

Reassuringly, we detected very few samples with predicted resistance to mefloquine, and none in Cambodia where this drug is now used as part of the first-line ACT. On multiple occasions since the 1990s, the use of mefloquine has intermittently led to reduced treatment efficacy in the GMS, due to the emergence of parasites possessing *mdr1* amplifications[39]. However, between 2019 and 2021 we found only 9 isolates with *mdr1* amplifications, all in the province of Attapeu in Laos; seven of these belonged to the *kelch13* R539T strain responsible for a *Pf* outbreak in 2021[32]. It is possible that this amplification was under selection in that population because it confers somewhat reduced sensitivity to lumefantrine, the partner drug in the frontline ACT of choice in Laos[40,41]. Our data showed a strong association between the *mdr1* NFD haplotype (N86, Y184F, D1246) and the ART-R

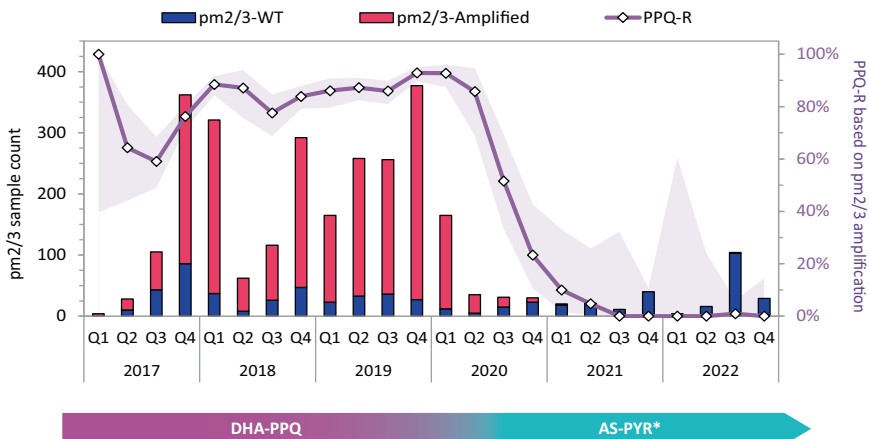

**Fig. 4 | Predicted levels of piperaquine resistance in five endemic provinces of Vietnam.** The bar chart shows the quarterly numbers of samples (left axis) with wild-type (WT, navy) and *plasmepsin2/3* gene amplification (red), against the derived proportion of piperaquine resistance (PPQ-R) samples in five endemic provinces in Vietnam (right axis). Purple shaded area shows 95% confidence interval of the PPQ-R proportion estimate. Q1: January–March; Q2: April–June; Q3:

July–September; Q4: October–December. The bar below the graph shows first-line treatment policy for uncomplicated *P. falciparum* in Vietnam, showing the timeline of transition from dihydroartemisinin-piperaquine (DHA-PPQ) to pyronaridine-artesunate (AS-PYR). *AS-PYR was adopted in 5 endemic provinces (Binh Phuoc, Dak Nong, Gia Lai, Dak Lak, and Phu Yen)[20,21]. Source data are provided as a Source Data file.

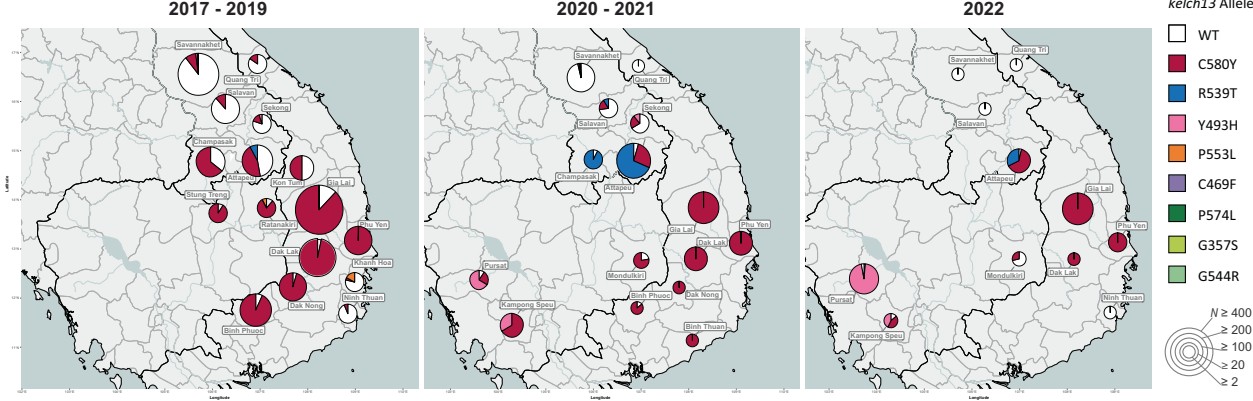

**Fig. 5 | Prevalence of *kelch13* alleles between 2017 and 2022.** The pie chart shows the proportions of *kelch13* alleles in each province where samples were collected. Pie chart size represents the number of samples from the province (N). Wild-type (WT) parasites are predicted to be sensitive to artemisinin. All but two of the *kelch13* alleles

detected in this study have been associated with delayed parasite clearance, and thus predictive of artemisinin resistance. Since the two rare alleles G357S and G544R are not in World Health Organization's validated marker list, their association with artemisinin resistance is undetermined. Source data are provided as a Source Data file.

*kelch13* variants (in particular C580Y, R539T, Y493H and C469F). Since this haplotype has been implicated in reducing susceptibility to antimalarials including piperaquine[42] and mefloquine[43], it is possible that it is under selection due to its effect on partner drug response.

Although we can predict resistance for a broad range of antimalarials, a limitation to the current study is the lack of validated genetic markers of resistance to a number of ACT partner drugs, including pyronaridine, used in Thailand and Vietnam, and lumefantrine, used in Laos. Pyronaridine is a relatively new drug, used only as a partner drug, for which no resistance has been detected to date; a recent meta-analysis of 5711 patients found that failure rates remain below 5%[44]. Lumefantrine, on the other hand, is used in oral co-formulations with artemether as first-line treatment in most African countries. While some studies have described phenotypic effects of *mdr1* variants, evidence is not strong, and *mdr1*-induced reductions of in vitro susceptibility have not been reconciled with in vivo clinical outcomes[45,46]. Further research, such as a highly statistically powered genome-wide association study (GWAS), is needed to identify markers

that can track resistance to these drugs, an urgent matter in view of the recent emergence of artemisinin-resistant parasites in eastern Africa[47].

Since GenRe-Mekong supports public health agencies, our protocols were designed to facilitate integration with routine healthcare in low-resource settings, which demands some trade-off. As a result, the present analysis is limited to symptomatic patients presenting at public health facilities, and excludes asymptomatic infections. We also acknowledge there are gaps in our coverage, due to differences in implementation schedules between countries, and a focus on the districts selected by NMCPs as having the greatest relevance. In spite of these limitations, we have shown that continued systematic routine surveillance with dense territory coverage is informative about epidemiological changes and allows comparisons of prevalence between regions and over time. Monitoring genetic markers allows efficient and rapid tracking of resistance, at a lower cost than therapeutic efficacy studies[48]. As the GMS moves closer to elimination, and the number of infected *Pf* patients dwindle, parasite populations will continue to change rapidly, and longitudinal surveillance will be an important tool

**Table 1 | Temporal distribution of *P. falciparum* sample clusters**

| Cluster | Previous names | Proportion of samples in each year | | | | | | Resistance | | | | | | Main kelch13 | KEL1/PLA1 |
|---|---|---|---|---|---|---|---|---|---|---|---|---|---|---|---|
| | | 2017 | 2018 | 2019 | 2020 | 2021 | 2022 | ART | PPQ | MQ | CQ | SX | PM | | |
| KLV01 | | 5% | 8% | 21% | 16% | 4% | 0% | 100% | 89% | 0% | 100% | 100% | 100% | C580Y | Yes |
| KLV02 | | 1% | 12% | 18% | 4% | 0% | 0% | 100% | 98% | 0% | 100% | 100% | 100% | C580Y | Yes |
| KLV03 | | 0% | 1% | 13% | 10% | 16% | 33% | 100% | 54% | 0% | 100% | 100% | 100% | C580Y | Yes |
| KLV04 | LAA1[a] | 0% | 0% | 0% | 18% | 15% | 4% | 100% | 0% | 13% | 100% | 100% | 100% | R539T | |
| KLV05 | | 4% | 4% | 3% | 3% | 0% | 0% | 100% | 78% | 0% | 100% | 100% | 100% | C580Y | Yes |
| KLV06 | | 5% | 3% | 1% | 1% | 0% | 0% | 6% | 1% | 0% | 100% | 0% | 100% | WT | |
| KLV07 | LAA2[a]; KH2[b] | 1% | 1% | 0% | 3% | 10% | 9% | 100% | 1% | 0% | 100% | 100% | 100% | C580Y | |
| KLV08 | | 5% | 2% | 0% | 0% | 0% | 0% | 100% | 84% | 0% | 100% | 100% | 100% | C580Y | Yes |
| KLV09 | | 1% | 1% | 3% | 4% | 0% | 0% | 100% | 100% | 0% | 100% | 100% | 100% | C580Y | Yes |
| KLV10 | | 3% | 2% | 0% | 0% | 0% | 0% | 1% | 0% | 0% | 100% | 0% | 100% | WT | |
| KLV11 | | 0% | 0% | 0% | 0% | 14% | 0% | 0% | 0% | 0% | 0% | 0% | 100% | WT | |
| KLV12 | | 0% | 0% | 0% | 0% | 1% | 16% | 100% | 0% | 0% | 100% | 100% | 100% | Y493H | |
| KLV13 | | 2% | 1% | 1% | 0% | 0% | 0% | 0% | 2% | 0% | 100% | 100% | 100% | WT | |
| KLV14 | | 0% | 1% | 2% | 1% | 0% | 0% | 100% | 85% | 0% | 100% | 100% | 100% | C580Y | Yes |
| KLV15 | | 0% | 2% | 0% | 0% | 0% | 0% | 0% | 0% | 0% | 100% | 100% | 100% | WT | |
| KLV16 | | 1% | 1% | 0% | 0% | 0% | 0% | 8% | 0% | 0% | 100% | 0% | 100% | WT | |
| KLV17 | | 0% | 1% | 0% | 3% | 0% | 0% | 0% | 0% | 0% | 100% | 0% | 100% | WT | |
| KLV18 | | 2% | 1% | 0% | 0% | 0% | 0% | 100% | 97% | 0% | 100% | 100% | 100% | C580Y | Yes |
| KLV19 | | 0% | 1% | 0% | 0% | 0% | 0% | 0% | 0% | 0% | 100% | 0% | 100% | WT | |
| KLV20 | | 0% | 1% | 0% | 0% | 0% | 0% | 0% | 0% | 0% | 100% | 0% | 100% | WT | |
| KLV21 | | 0% | 1% | 1% | 1% | 4% | 1% | 0% | 0% | 0% | 100% | 100% | 100% | WT | |
| KLV22 | | 1% | 0% | 0% | 0% | 0% | 0% | 100% | 0% | 0% | 100% | 100% | 100% | WT | |
| KLV23 | | 0% | 1% | 0% | 0% | 0% | 0% | 100% | 100% | 0% | 100% | 100% | 100% | C580Y | |
| KLV24 | | 1% | 1% | 0% | 0% | 0% | 0% | 100% | 0% | 0% | 100% | 100% | 100% | C580Y | Yes |
| KLV25 | | 0% | 1% | 1% | 1% | 0% | 0% | 75% | 0% | 0% | 100% | 100% | 100% | C580Y | |
| KLV26 | | 0% | 0% | 1% | 1% | 0% | 0% | 100% | 94% | 0% | 100% | 100% | 100% | C580Y | Yes |
| KLV27 | | 1% | 0% | 0% | 0% | 0% | 0% | 0% | 7% | 0% | 100% | 100% | 100% | WT | |
| Not clustered | | 55% | 43% | 30% | 26% | 26% | 36% | | | | | | | | |
| Total samples | | 1291 | 1894 | 1447 | 608 | 413 | 329 | | | | | | | | |

For each cluster, we show: the cluster label; the name by which the cluster was identified in previous studies, if any; in each year of collection, the proportion of samples that belong to the cluster; the proportion of samples predicted resistant to artemisinin (ART), piperaquine (PPQ), and mefloquine (MQ), chloroquine (CQ), sulfadoxine (SX) and pyrimethamine (PM); the majority *kelch13* variant carried by parasites in the cluster; and whether the cluster carries a KEL1/PLA1 haplotype. Only clusters with ≥20 members are shown; the full list of clusters can be found at Supplementary Table 5. Samples that were not assigned to a cluster with these parameters were labeled as 'not clustered'.
[a]Wasakul et al.[32].
[b]Miotto et al.[34].

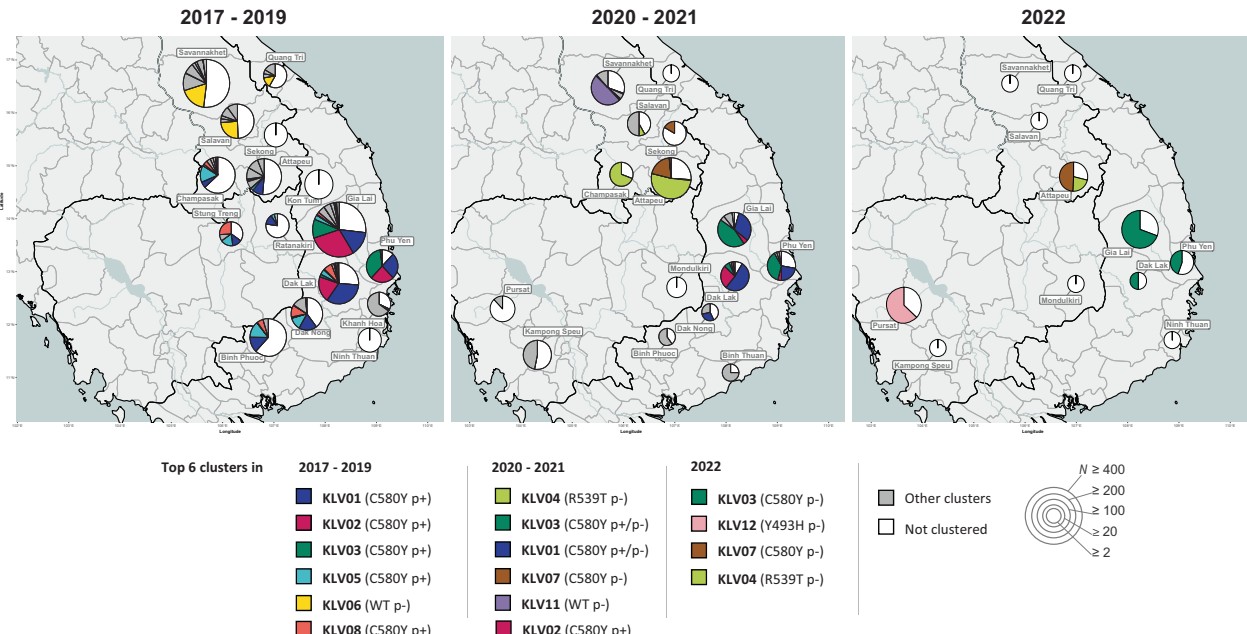

**Fig. 6 | Proportion of the most prevalent *P. falciparum* clusters in each province across three periods.** Left panel: 2017–2019, middle: 2020–2021, right: January–December 2022. Clusters were identified by applying community detection algorithm to a graph of parasites sharing at least 95% genetic barcode identity, identifying 64 clusters with at least 10 members. In 2017–2019, 54 clusters were present, while 14 clusters were present in 2020–2021 and only four clusters were found in 2022. Pie chart size represents the sample size for the province (*N*). To improve visualization, the top six clusters in each period were assigned colors, while the remaining smaller clusters are shown in gray. White segments represent the proportions of samples that were not assigned to a cluster. In the legend, clusters are arranged in descending order of cluster size for each period. The main *kelch13* variant observed in the cluster and a label denoting *plasmepsin2/3* amplification (p+) or *plasmepsin2/3* wild-type (p−), are shown in brackets following the cluster name. Source data are provided as a Source Data file.

for supporting agile interventions. At the same time, surveillance planning will become increasingly challenging, given the difficulty of predicting the smaller outbreaks that can be expected in the elimination phase. In this epidemiological scenario, genetic surveillance will focus on use cases relevant to sustained elimination and prevention of reintroduction, such as detection of imported foreign malaria infections and outbreak analyses[11,49]. Furthermore, as *Pf* incidence diminishes, NMCPs will have to control other *Plasmodium* species, notably *P. vivax*, for which very few surveillance tools are currently available. GenRe-Mekong is committed to supporting NMCPs in the GMS with genetic surveillance tools for *P. vivax*, currently under development.

One of the key challenges of genetic epidemiology is that of translating genetic data into actionable knowledge relevant to NMCP activities, since mutations and haplotypes have little value without an understandable context in which they can be assessed[11]. We developed the grcMalaria software library with the aim of increasing the accessibility of genetic epidemiology analysis in daily practice, simplifying and reducing time needed for analyses tasks. This freely available tool focuses on mapping, using administrative divisions that are meaningful to NMCPs, and supporting flexible selection of samples and division of the dataset into time slices, producing parallel maps for comparison. The library incorporates clustering algorithms that group samples by similarity and map the distribution of these clusters, allowing users to distinguish local strains from those spreading across borders.

The results presented in this study emphasize the importance of regional strategies for selection of treatment policies to mitigate the spread of antimalarial drug resistance, and stresses the value of selecting different first-line treatment regimens in neighboring countries. For future malaria elimination efforts, genetic surveillance will support public health authorities, informing strategic planning in the final phases of elimination.

## Methods

The procedures, operations and technical aspects of the GenRe-Mekong project were detailed in a previous publication[9]; here, we briefly summarize the salient points.

### Sample collection

Samples were collected from symptomatic patients of all ages who had been confirmed positive for *Pf* by positive rapid diagnostic test or blood smear microscopy. Written informed consent was obtained from each participant, or from parent/guardian or legally authorized representative, with patient assent provided where required by national regulations. Following consent and prior to treatment, two or three 20 μL DBS were collected on filter paper by finger-prick, labeled with a unique anonymous identifier, and stored in plastic bags with silica gel before being shipped to the laboratories. In Laos, samples were collected in five provinces by the Center for Malariology, Parasitology, and Entomology (CMPE) and the Lao-Oxford-Mahosot Hospital-Wellcome Trust Research Unit (LOMWRU); in Vietnam, sampling was carried out from 11 provinces by the Institute of Malariology Parasitology and Entomology Quy Nhon (IMPE-QN), the National Institute of Malariology, Parasitology and Entomology (NIMPE) and the Oxford University Clinical Research Unit (OUCRU); and in Cambodia, samples were collected in five provinces by the National Center for Parasitology, Entomology and Malaria Control (CNM) (Supplementary Table 2). In each country, the sampling networks were planned by the NMCP, who selected and enrolled sites in collaboration with the relevant provincial health authorities. Staff training was delivered jointly by NMCP officers, provincial health officers and GenRe-Mekong staff. Ethical approvals were obtained both from ethics committees in the country of collection, and from the Oxford Tropical Research Ethics Committee (OxTREC), as previously detailed[9].

## Sample genotyping

DNA was extracted from DBS samples and processed using the Spot-Malaria v2 amplicon sequencing protocol and primers, as previously detailed[9]. Processing took place in GenRe-Mekong partner laboratories at OUCRU, Ho Chi Minh City, Vietnam; or at the Wellcome Sanger Institute (WSI), Hinxton, UK. Supplementary amplification genotyping by qPCR was carried out by OUCRU. The data from sequencing and qPCR were subsequently used to call genotypes for a standardized set of genotypes, gene amplifications and haplotypes associated with resistance to several antimalarials; to construct genetic barcodes consisting of 101 SNP selected for their ability to differentiate populations and their power to recapitulate genetic distance; and to predict the drug resistance status of each sample. Details of the genotyping pipeline and procedures are given in the Supplementary Methods; a list of the drug resistance-related variants genotyped is provided in Supplementary Table 1.

## Data analysis and visualization

To analyze and visualize data from the GRC, we used the grcMalaria package (version 2.0.0)[50] running on R (version 4.2.3)[51]. The GRC format requires structured geographical data, which is achieved by linking study locations to administrative divisions from GADM (Database of Global Administrative Areas)[52], a global catalog of administrative divisions and their coordinates. The grcMalaria package processes standardized genetic surveillance data in the form of a GRC data file, a Microsoft Excel spreadsheet formatted according to the GenRe-Mekong GRC Data Dictionary (https://github.com/GenRe-Mekong/Documents/tree/main/GRC-DataDictionary). The grcMalaria package is not limited to analyzing GMS surveillance data presented in this study but is capable of generating maps across different geographical regions using data in the GRC format, such as the global dataset from the MalariaGEN Pf7[53].

Dedicated functions generate genetic epidemiology maps (including sample distribution, mutation prevalence, and drug resistance maps) and their associated data, which are delivered as image and spreadsheet files. Geographical structuring supports the aggregation of data at different levels of geographical granularity. In all analyses presented here, we aggregated samples at provincial level (First-level Administrative Division). To analyze Pf population changes over time, we partitioned our sample set into three periods (2017–2019, 2020–2021, 2022) characterized by different epidemiological scenarios. Sample distributions were mapped using the grcMalaria mapSampleCounts function. Malaria prevalence was calculated as the proportion of confirmed Pf cases reported to WHO over total number of tested cases[28–30]. To visualize the frequency of parasites resistant to a given antimalarial, we estimated resistance prevalence using the grcMalaria function mapDrugResistancePrevalence, which calculates the proportion of samples predicted as resistant, disregarding samples whose drug-resistant status could not be predicted (e.g., mixed infections and low-quality samples). Pie charts, representing allele proportions of drug-resistant markers by administrative division, were generated using the grcMalaria function mapAlleleProportions. Allele proportions at a given position were calculated by dividing the number of samples carrying each allele by the total number of samples in the administrative division, disregarding samples carrying multiple alleles and those missing a genotype at that position.

Seven loci of the chloroquine resistance transporter (crt) gene are associated with piperaquine resistance (PPQ-R)[31]. Four of these loci, including T93S, H97Y, I218F, G353V, were genotyped in the Spot-Malaria GRC using amplicon sequencing. Of the 5982 samples presented in the manuscript, 3066 had available genotypes at these positions. To obtain the remaining PPQ-R associated crt mutations, we genotyped the relevant positions using whole-genome sequencing data, available for a large subset of samples (n = 1581), thus identifying additional genotypes at crt F145I and M343L. Due to the technical complexity of crt genotyping, a proportion of samples yielded incomplete or missing genotype data. Samples lacking genotype data at any of these loci were excluded from the frequency calculation. In our sample set, no genetic variations were detected at position 350 of the crt gene, and therefore we reported no analysis results for the WHO-listed mutation C350R.

Graphs were produced either by Microsoft Excel or by the R ggplot2 (version 3.5.1) library[54]; error bars and shaded areas representing 95% confidence intervals were calculated using Wilson Interval with corrections (BinomCI; DescTools R package version 0.99.54)[55].

## Statistics

A Cochran-Armitage trend test was conducted to evaluate trends in the proportion of resistant parasites across different periods (CochranArmitageTest; DescTools R package version 0.99.54). Bonferroni post-hoc analyses between group pairs were performed to adjust for the increased risk of Type I errors. Segmented regression analysis was applied to evaluate the impact of two policy changes (frontline ACT switch and COVID-19 movement restrictions) on malaria prevalence and frequencies of pm23 amplifications over time. Breakpoints corresponding to each intervention were identified using the segmented package in R, allowing estimation of shifts in trend before and after each intervention while accounting for time-dependent effects. Where segmented regression was not applicable, due to the absence of policy changes, linear regression was applied to assess general trends over time. Spearman's rank correlation coefficient was used to assess relationships between non-normally distributed malaria prevalence and PPQ-R frequencies (corr.test; stats R package version 4.2.3). Logistic regression with kelch13 alleles as the dependent variable was used to evaluate association with mdr1 haplotypes (glm; stats R package version 4.2.3).

## Analysis of highly related parasites

Clusters of highly related parasites were identified based on pairwise genetic barcode similarity s, obtained from the pairwise genetic distance d where $s = (1-d)$. Pairwise genetic distances between two samples were computed by comparing the alleles carried by the two samples at each of the barcoding SNPs, as previously described[32]. To minimize the influence of missingness on genetic distance estimations, we only included high-quality samples (<25% barcode genotype missingness), and high-quality barcoding SNPs (<20% samples with missing genotype), and applied a simple imputation method at positions where the genotype was missing. At each missing genotype, the imputation method used the pairwise genetic distance matrix to identified the 100 samples closest to the sample being imputed, and assigned the most common allele observed in this group as the imputed allele. Cluster detection was performed by the grcMalaria findClusters function, which constructs a graph connecting sample pairs with s greater than a specified minimal threshold $s_{min}$, and then applies the Louvain multilevel community detection algorithm[56]. We specified a genetic similarity threshold of 95% ($s_{min} = 0.95$) to cluster parasites with essentially identical genetic barcodes, while allowing for low levels of genotyping error; clusters with fewer than 10 members were disregarded. The proportion of the most prevalent clusters were visualized as pie charts, generated using the grcMalaria mapClusterSharing function.

## Data availability

This paper uses a subset of GenRe-Mekong data, which includes all samples collected by the project during routine surveillance in the eastern GMS. The GenRe-Mekong project has made all genetic data and its corresponding metadata openly available at https://github.com/GenRe-Mekong/Data, including: the full GenRe-Mekong dataset, containing 13,647 samples from eight countries; a subset that includes 5982 Pf samples from the present analysis; and a global dataset containing 16,203 high quality samples, derived from whole-genome sequencing data from the MalariaGEN Pf7 data resource[53]. The datasets are provided in the GRC format, along with a data dictionary for converting third-

party data into GRC format for compatibility with the grcMalaria v2.0.0 library. Sequence data for this study has been deposited in the European Nucleotide Archive at EMBL-EBI under accession numbers PRJEB85801 and PRJEB2136. Source data are provided with this paper.

## Code availability

Links to source code for the grcMalaria R package[50] can be found at https://github.com/GenRe-Mekong/grcMalaria, and a comprehensive guide detailing its functions is available at https://genremekong.org/tools/grcmalaria-guide. For quick exploration of published data, an interactive web tool built on the grcMalaria R package is available at https://genremekong.org/tools/grcmapper.

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

## Acknowledgements

This study was funded by the Bill & Melinda Gates Foundation (grant numbers OPP11188166, OPP1204628, INV-001927) and by The Global Fund to Fight AIDS, Tuberculosis and Malaria (grant number QSE-M-UNOPS-MORU-20864-007-44) granted to O.M. This research was funded in part by the Wellcome Trust (grant number 204911 to Prof. Dominic Kwiatkowski). The authors wish to thank all the patients and guardians who generously agreed to provide blood samples, and all public health and medical staff who participated in the collection of those samples. Genome sequencing and genotyping was performed by the WSI and OUCRU, and sequencing data processing was supported by the MalariaGEN Resource Centre. We thank the staff of the WSI Sample Logistics, Sequencing, and Informatics facilities for their contribution; Eleanor Drury for support in the sample processing pipeline.

## Author contributions

V.W., T.D.V., and O.M. conceived and designed the study. N.T.-N., H.H.Q., N.D.T., K.C., S.S., S.M., V.B., V.S., M.M., N.T.K.T., V.N.L.P., S.C., S.P., and S.X. coordinated sample and data collection. P.D.T., N.T.-N., S.G., R.D.P., and C.V.A. processed samples, and generated genomic data. V.W., T.D.V., E.B., and O.M. developed methods, analyzed and interpreted data. V.J.S., P.N.N., E.A.A., R.J.M., N.P.D., A.M.D., and O.M. managed and coordinated the genetic surveillance project. V.W., T.D.V., E.B., and O.M. wrote and revised the manuscript. V.W., T.D.V., and O.M. accessed and verified all the data. All authors provided critical revision of the manuscript. All authors had full access to all the data in the study and had final responsibility for the decision to submit for publication.

## Competing interests

The authors declare no competing interests.

## Additional information

[1]Mahidol Oxford Tropical Medicine Research Unit, Faculty of Tropical Medicine, Mahidol University, Bangkok, Thailand. [2]Oxford University Clinical Research Unit, Ho Chi Minh City, Vietnam. [3]Institute of Malariology, Parasitology and Entomology, Quy Nhon, Vietnam. [4]National Institute of Malariology, Parasitology and Entomology, Hanoi, Vietnam. [5]Centre for Malariology, Parasitology and Entomology, Vientiane, Lao PDR. [6]National Center for Parasitology, Entomology and Malaria Control, Phnom Penh, Cambodia. [7]Lao-Oxford-Mahosot Hospital-Wellcome Trust Research Unit, Microbiology Laboratory, Mahosot Hospital, Vientiane, Lao PDR. [8]Institute of Research and Education Development, University of Health Sciences, Ministry of Health, Vientiane, Lao PDR. [9]Centre for Tropical Medicine and Global Health, Nuffield Department of Medicine, University of Oxford, Oxford, UK. [10]Saw Swee Hock School of Public Health, National University of Singapore, Singapore, Singapore. [11]Wellcome Sanger Institute, Hinxton, UK. [12]The Open University, Milton Keynes, UK. ✉e-mail: olivo@tropmedres.ac

