## [Peer Review file · Nature Communications]

Genetic surveillance of *Plasmodium falciparum* populations following treatment policy revisions in the Greater Mekong Subregion

Corresponding Author: Dr Olivo Miotto

Version 0:

Reviewer comments:

Reviewer #1

(Remarks to the Author)

Review GENREMEKONG paper

The manuscript describes genetic surveillance of drug resistance of *Plasmodium falciparum* in the Greater Mekong Subregion, and presents promising results. It shows a remarkable decrease in plasmepsin 2/3 amplifications, from a prevalent 62% to a mere 2%. While artemisinin resistance remains high, the absence of resistance to partner drug mefloquine in Cambodia is encouraging.

Strengths:

The study includes a large number (5982) of dried blood spot samples from multiple provinces in 3 different countries from 2017-2022. The authors applied a well-standardized genotyping protocol, both in the lab and in silico, which have been made openly available. Results can have important implications for the field.

Major Concerns:

1. Conclusions on DHA-PPQ Resistance: The strong conclusions regarding the disappearance of DHA-PPQ resistance in the GMS are concerning. The authors based their predictions of PPQ resistance on plasmepsin 2/3 amplifications alone, disregarding strong evidence from the same region of emerging mutations in the crt gene, which are also validated markers of PPQ resistance according to the WHO.

The evidence for crt mutations includes a study from this journal that demonstrated that mutations in crt emerging in Cambodia caused in vitro resistance to PPQ, also without the presence of the plasmepsin 2/3 amplification (Ross et al. Nat Comms 2018). Another clinical study published in the Lancet Infectious diseases journal, including co-authors from the current study, confirmed the association of these crt mutations with DHA-PPQ failure in Cambodia (van der Pluijm et al 2019, Lancet Inf Dis). While in the clinical study the crt mutations were seen in KEL1/PLA strains with pm2/3 amplifications, from the loss of these amplifications alone in the current study, the authors cannot conclude that the parasites are susceptible.

Without genotyping data of this region of the crt gene and the lack of phenotypic characterization of the strains without pm2/3 amplifications, it cannot be concluded that DHA-PPQ resistance is no longer present in the studied area. The conclusions made by the authors could lead to misinterpretations with severe consequences for clinical practice.

It would be valid to conclude that there is resistance in an area where you encounter only one of multiple resistance markers. It is however very risky to conclude the return to susceptibility based on the absence of only one of multiple validated makers.

2. Presentation of Data: The way the data is presented could lead to misinterpretation of the data. For example, the multipanel figures showing pie charts that are proportional for the sample sizes do not clearly indicate the differences in sample sizes between the periods.

The size of pie-charts is only proportional to the sample size for the province within the time period, and it is not proportional

between the different panels (i.e. different periods). Due to the big differences in sample size this should be improved.

3. Generalization of Findings: The manuscript makes associations between treatment policy changes and changing population dynamics, while the data does not strongly support this statement. The treatment policy only changed in Vietnam during the study period, while the populations in that country did not change (based on the barcode data). Findings in one country should not be so strongly generalized for the region, especially considering the fragmented sample collections and differing results and treatment policies in the three countries in this study.

4. Influence of External Factors: The influence of external factors, such as restrictions during the COVID-19 pandemic, on malaria incidence, sample collection, and observed patterns is not fully addressed.

In conclusion, more nuance is needed in presenting and interpreting the data throughout the manuscript.

Specific major comments

- Line 122-123 - The groups are not very even this way. . It might make more sense to make two groups: 2017-2019 vs. 2020-2022. Any specific trends in 2022 you would want to highlight, you could do so in the text.
 - o This way also the difference in sample size will become more clear in these figures. For example in 2022, the size of the piechart from Attapeu (59 samples) is only very slightly larger than Quang Tri with only 3 samples. In 2017-2019 the size of the piechart of Quang tri is the same as in the later years, but in 2017-2019 you had 64 samples for this province, while in 2020-2021 there were only 2 and in 2022 there were only 3 samples.
- Line 165 – Figure 1A does not show the incidence of malaria in Vietnam; it shows the number of samples collected. From your samples you probably cannot calculate/estimate incidence and you will need to refer here only to the data reported to the WHO.
- Line 168 – “As incidence declined over time, Pf populations became more geographically isolated.” Based on what results/outcome variable is this statement based? It is not clear on the basis of the results presented so far.
- Lines 172-191 and Figure 2 and Figure 3 => PPQ resistance was predicted only on the basis of PM2/3 amplifications. The manuscript should clearly state that other markers of PPQ resistance were not investigated.
- Lines 177-181 “Although KEL1/PLA1 began to diversify as it spread, it maintained its two characteristic 178 haplotypes during this period: a kelch13 C580Y mutant haplotype conferring artemisinin resistance, 179 and a pm23 amplification conferring resistance to piperazine. This strain also spread into southern Laos, likely as a spillover from neighboring countries, producing high levels of DHA-PPQ resistance in 181 Attapeu and Champasak Provinces, despite the ACT not being used nationally.” => where are these results? It is not clear to me from table 1 or figure 5 that the lineages in Laos were a spillover from neighboring countries?
- Line 183-185 “Following these changes, 184 regional DHA-PPQ predicted resistance fell steeply, to 30% (204/690) in 2020-2021, and 1% (2/278) in 2022 (Figure 2a, Supplementary Figure 3a).” => Please add statistics to support this statement, especially considering the huge difference in sample sizes: 62% (1,957/3,132) to 30% (204/690) to 1% (2/278). You can also see that the confidence interval in figure 3 for the ‘PPQ-R’ in 2021 and 2022 is quite wide.
- Line 185-188: “To demonstrate that these changes in first-line treatment policies were the likely cause of DHA-PPQ resistance decline, we plotted the prevalence of predicted piperazine resistance (i.e. the pm23 amplification prevalence), alongside the timeline of first-line treatment strategies for uncomplicated Pf in Vietnam (Figure 3).” I am not sure that this is the right conclusion to make, it is circumstantial at best.
 - o There are several issues here:
 - When plotting the pm2/pm3 amplifications, I would only do so for the provinces where the treatment policy has actually changed.
 - I would change the legend of the blue line to better reflect that this is only based on PM2/3 and not all validated markers.
 - Why only analyze this for Vietnam? And not for the other countries?
 - As far as I am aware Pyramax was already used prior to 2020 in several of these provinces in Vietnam. Dak Nong and Binh Phuoc changed to pyramax as first line treatment mid-2019, after results from TES trials with pyramax were analyzed (conducted in sentinel sites in several provinces in 2017 and 2018). In Dak Lak and Gia Lai treatments changed early 2020.
 - There were a lot of other things going on in 2020/2021 in Vietnam that have potentially impacted malaria transmission and sample collections, including extensive strict limitations to cross-border travel between provinces due to COVID outbreaks and lock downs preventing people to leave the house for example to go into the forest or fields where they are more at risk of infection.
- Line 188-191: “We observed that within 3 months after replacing DHA-PPQ with AS-PYR in 5 provinces, a reduction in Pf incidence and a decline in PPQ 190 resistance began, which within 18 months led to the disappearance of piperazine-resistant parasites 191 (Figure 3).”
 - o Same comments as raised before: nr of samples collected is not the same as incidence. Especially considering barriers to patients seeking access to healthcare and barriers to sample collections during restrictions as a result of COVID.
 - o AS-PYR was not implemented at the same time in all provinces.
 - o PPQ resistance based on PM2/3 alone
- Line 194-195: “presumably because artemisinin-based combinations continued to be the first-line treatments in the region.” Avoid presenting speculations in the results section.
- Line 195-199: “However, in provinces at the border of the endemic region (Savannakhet in Laos, Quang Tri and Ninh Thuan in Vietnam), most parasites were predicted to remain susceptible to both artemisinin and piperazine throughout the study period (Figure 2b), suggesting that the local parasite populations were isolated from the spreading resistant strains in the eastern GMS.” I am not sure that Ninh Thuan qualifies as “the border of the endemic region”, and that this is the reason why the parasites in this region are different. The coastal provinces have historically had different malaria populations and factors driving transmission than the Central Highland provinces, through a complex interplay of environmental and socio-economical factors. There are probably other factors at play both here and probably in other provinces as well that contribute to the genetic patterns observed rather than just being at the edges of the (current) endemic regions.
- Line 218: It is interesting that C580Y remained the dominant haplotype in Vietnam, while both in Laos and Cambodia this really changed.

- Lines 227-232: "Following the implementation of new front-line treatment policies, however, a noticeable shift in population dynamics occurred. In Vietnam, the KEL1/PLA1-derived clusters KLV01 and 232 KLV03 continued to circulate but lost their pm23 amplification."

o The statement regarding the shift in population dynamics following the implementation of new front-line treatment policies is not strongly supported by the data.

o The treatment policies only changed in Vietnam in the study period, while the population/the barcode in the Vietnam samples did not really change, but this strain 'just' lost its PM2/3 amplifications and became predominant. This indicates evolution of this lineage, not a shift in population dynamics.

o In Cambodia, treatment policy already changed in 2017, but the population in Pursat changed in 2022 compared to 2020-2021. There are no samples from Pursat from 2017 to 2019, in addition, the sample size in 2020-2021 is much lower than in 2022 in this province. The evidence for a population shift directly due to the change in treatment policies is weak.

o In Laos, the treatment policies have remained the same throughout the entire study period, so a change in treatment policies cannot be the explanation for the emerging(?) KLV04 and KLV07 lineages.

- Line 232- 234: "By the third 233 quarter of 2021, none of the circulating clusters possessed the pm23 amplification, and the Central 234 Highlands of Vietnam were dominated by the piperazine-susceptible KLV03 population (Figure 5)." – is there any clinical or phenotypic data to support that the KLV03 population is piperazine-susceptible? The absence of pm2/3 amplifications is not sufficient for this conclusion.

- Line 235-237: "Taken together, these results suggest that the observed reduction in DHA-PPQ resistance in 2020- 236 2021 can be explained by the disappearance of KEL1/PLA1 clusters, and their loss of pm23 237 amplification, occurring after changes in first-line treatment policies."

o There is no reduction in DHA resistance => still very high levels of K13 associated mutations.

o KEL1/PLA1 clusters have not disappeared in Vietnam, they have changed. The clusters are identified based on the barcode, which did not change, but they have lost the pm2/3 amplification.

o The data is incomplete to suggest that there is no more PPQ resistance. Genotypic data for crt alleles associated with PPQ resistance are missing, nor is there any phenotypic or clinical data to support this.

o Clusters have changed in Cambodia and Laos, but the link to changed treatment policies is not that clear (see earlier comment).

- Line 248-249: "Our data shows that dramatic epidemiological changes occurred following changes in first-line treatment policies in the region." – The study design is not appropriate to determine epidemiological changes.

- The discussion should be adjusted based on the requested nuances and changes in interpretation highlighted in the comments to the results section and main concerns section above.

Minor comments:

- Ensure the number of provinces in Vietnam is consistent throughout the manuscript.

o Where are the results from Quang Nam province?

- Address the limitations in sample collection and the potential impact on the results.

o Sample collection was limited to specific provinces, and not all provinces sampled in all years. potentially missing variations in other regions and years.

o The study relies on samples from symptomatic patients, which may -not represent the entire population, including asymptomatic carriers.

- Line 106-109 – please provide references or details for the different PCRs used.

- Line 127-129 – Clarify how allele proportions were determined and how multiple clone infections were taken into account.

- Line 135-138: Include details on the Louvain algorithm in the methods section.

o How are missing positions imputed? Is any other data used for the imputation, for example, is imputation done at each province level?

o How does your algorithm with at least 20 members impact the interpretation of the data from 2022, where you do not have 20 samples from many provinces. You mention somewhere in the results that the provinces are becoming isolated, but this is difficult to conclude as a large part of your samples in 2020-2021 and 2022 do not belong to a cluster. Considering the rules you applied for the clustering, with such low numbers of samples, can the absence of clustering be interpreted as increased isolation?

- Lines 238-244: these results are very separate from the main aims of the manuscript and it is unclear why they are added here. If it to present and demonstrate the software-tool, then the development of this software tool needs to be better described in the methods. What does this tool do and why is it different from the tool used for example in the Jacob et al. paper.

(Remarks on code availability)

An R package is available on github and a userguide and tutorial is available on <https://genremekong.org/tools/grcmalaria-r-package-user-guide>.

The package is very useful to the community and allow replication of the analysis. In addition, the software package can be used to analyse your own Genetic Report Cards.

I tested the package by downloading and running parts of the example notebook.

Reviewer #2

(Remarks to the Author)

Verschuuren et al. summarise P.f. drug resistance genotypes from ca. 6000 samples collected across the Greater Mekong Subregion (GMS) from 2017 – 2022. The genotyping data is generated using a previously published protocol & bioinformatic

pipeline, and visualisations are made using a spatial mapping software (grcMalaria) the authors developed in R and have now made public.

The manuscript adds to a growing number of studies where genomic surveillance of P.f. is being used to aid and analyse policy decisions, and deploys one of now at least three existing multiplexed amplicon sequencing assays for Illumina. The paper continues from previous work where the authors sequenced ca. 10000 samples across the GMS using the same approach (<https://elifesciences.org/articles/62997#content>). Compared to related literature the work benefits from scale (e.g. more samples and geographical coverage) and a greater commitment to making data interpretable for policy makers through conversion to phenotype and visual aids. In my opinion, SpotMalaria/GenRe-Mekong currently represents the most complete effort to systematically realise the translational value of P.f. genomic data in policy.

That being said, from a methods point of view, I found no improvements in this manuscript over previous work. In addition, I believe the bioinformatics pipelines may now be outdated, and I was unable to find key validation experiments or sequencing quality control data (in this or previous manuscripts). Moreover, my opinion is the grcMalaria software is unlikely to be used by the wider community due to its complex input requirements.

The main value of the paper is epidemiological: in the presentation of longitudinal resistance trends in response to switching first-line treatments across the GMS. The authors make some interesting / important observations, but very similar observations have been made previously by an overlapping team (<https://doi.org/10.1128/aac.01121-21>), and I feel in this manuscript they need to be more thoroughly analysed and contextualised.

Major Points

1. The main finding from the manuscript is that "...dramatic epidemiological changes occurred following changes in first-line treatment policies across the region." (Lines 247 – 248). In my view there were two major potentially relevant policy changes: switching ACTs and COVID19 restrictions; and two dramatic observations: declining prevalence and PPQ resistance. While I appreciate that establishing causal links is challenging, the data supports much more investigation into these relationships than has been presented. For example, Fig 3 presents data only for Vietnam; Cambodia and Laos underwent different policy changes (earlier switch to AS-MQ; or retaining AL) and also investigating prevalence/PPQ-R trends here (potentially incorporating your prior data for a longer timeline) could strengthen evidence for first-line treatment being a causative factor (especially given the increased geographical isolation you observe might imply limited parasite migration). Relatedly, note that Lines 62 – 63 vs. Lines 181 – 182 give a contradictory timeline for the policy change in Cambodia. I also suggest that annotating Fig 3 with relevant COVID19 restrictions would help readers more intelligently interpret your main findings. At present, I believe more analysis can be done to support the main claim, and to try and disentangle COVID19/ACT policy effects (which the authors only mention briefly Lines 268 – 270).

2. A potentially high-value output is the grcMalaria package which was used to generate the spatial maps of resistance in the paper. I reviewed the github repo and commend the authors on the high code quality and extensive documentation. However, I highlight that the utility of grcMalaria to the scientific community is severely limited by the fact that it requires a very complicated-looking and non-standard input data format (i.e. the Genetic Report Card) that is produced by their upstream pipelines. Despite providing a description of the columns, practically speaking it would be very laborious / error-prone to convert from standard genotype output formats (e.g. VCF) into this excel spreadsheet. To rectify this, the authors need to support standard / simple input file types (VCF, a CSV with sample ID and collection location); e.g. by providing an R function to convert from standard input file types to GRC format. Furthermore, could you comment on how grcMalaria would work with amplicon sequencing data from other assays? As it stands, I believe grcMalaria will see little use by the wider community.

3. In the Methods, the authors cite their previous publication (<https://elifesciences.org/articles/62997#content>) and state they only summarise salient points in this manuscript. However, several salient points are missing. Critically, I can't determine clearly how PPQ resistance was called, despite that being central to the manuscript's conclusions. The methods say that qPCR was "also used" to detect amplification, but no specific qPCR assay is cited. The primary detection was presumably by amplicon sequencing, but in the technical notes of the previous paper for 'Piperazine' they mention targets against a breakpoint fusion amplicon in pm23 (which would be impressive if it worked, and would certainly require some validation) as well as novel targets in pfCRT and pfExo. How was data from these various targets and assays combined to make a final prediction of PPQ resistance? This needs to be clearly described in the main methods. Also, a basic description of the bioinformatic pipeline used to process all the sequencing data should be included.

4. The authors processed ca. 6000 samples of which ca. 900 were excluded due to high missingness. Aside from these numbers, the authors provide no data at all on sequencing performance or quality control. For example, it is standard practice to provide data on depth of sequencing across samples and amplicons (see Fig. 1 <https://www.nature.com/articles/s41467-024-46535-x>, Fig. 1 <https://www.nature.com/articles/s41467-023-39417-1>, or Fig. 2/3 <https://www.nature.com/articles/s41467-024-45688-z> as recent examples from this journal). Currently, readers have no way of evaluating the quality of the sequencing data that was generated, but rather have to take the final genotype / phenotype predictions on trust. Sequencing QC data should be included in the supplementary materials.

5. From the technical notes of the previous publication I believe that the genotyping is performed with bcftools. For Illumina amplicon sequencing data there are now many tools I understand to be more suitable / accurate for the variable ploidy seen in P.f. infections (DADA2, SeekDeep, HaplotypR). Is there a particular reason that bcftools is being used over these other methods, given that it (incorrectly, in this case) assumes a diploid genome with alleles at equal frequencies? I could not find

any validation of its performance on polyclonal infections in the previous manuscript, or an indication of rates of polyclonal infection in this dataset. My impression is that it will incorrectly call samples as homozygous if the minor allele within the sample is ca. <20%. I currently can't assess what impact this might have on overall results.

6. In several places throughout the manuscript (e.g. Line 164 – 165 and Fig 1; Fig 3) the number of samples collected (or passing QC) seems to be used as a proxy for P.f. clinical incidence / prevalence. While I understand you are sequencing a considerable percentage of all cases and there is a correlation here, this approach introduces unnecessary noise / confounding. Please use standard measures (prevalence / clinical incidence) when making statements / display items about transmission intensity.

Minor comments

Figure 1a: y-axis label is missing from. Moreover, adding the longitudinal prevalence trend to this figure would improve it, as it is a very important piece of contextual information that explains the reduce sample collection numbers.

Figure 2: PPQ-R should also be shown. Can be sympatric circulation of ART and PPQ in same site, as is one cannot infer pattern of PPQ-R resistance. I don't see PPQ-R spatial map in supplementary either but this is the key genetic change.

Supp. Fig 3a: I think this is panel communicates some of your most interesting results very nicely (esp. if you include DHA-PPQ as well). Consider incorporating in main text.

Lines 238 – 244: This section (Use of grcMalaria for non-GMS data) should really not be in the main results text in my opinion. The fact that grcMalaria can also make plots for other geographies is a detail of software capabilities, not a result. It could be added to the github repository / documentation, or to the 'Data analysis and visualisation' section of methods.

Lines 296 – 298: The claim is made that the results generated by the authors approach (e.g. local sample collection and DNA extraction, shipping internationally for sequencing and data analysis) was more efficient, cheaper, and reduced turn-around time relative to TES. While in principle these things can be true, in practice they can be very hard to realise. Can you support these claims with evidence, e.g. do you have numerical data on costs and turn-around times for the samples included in the study? Including data to support these claims in a supplementary figure would strengthen the paper; without it you can only claim this in principle.

Line 143, Data availability: Following links here I found one GRC (v1.2) with samples going only to 2021. Is the data as described from Lines 143 – 152 currently publicly available (at <https://github.com/GenRe-Mekong/Data>)? If not, when will it be made public?

Line 34/58: Here you state DHA-PPQ was previously dominant in eastern GMS, but my understanding from previous work was that it is primarily found in Cambodia or lower GMS? E.g. Figure 3 of <https://elifesciences.org/articles/62997#content> : the two coastal sites you have in Vietnam, and several sites in Laos and at the same longitude as DHA-PPQ-R populations in Cambodia are sensitive. Very minor point but good to keep a consistent / accurate story around location.

COI data: I believe COI is being estimated by the bioinformatic pipeline using THEREALMcCOIL. It would be interesting to many in the community to see if there was a decline in fraction polyclonal or mean COI following prevalence decline around GMS. Could be a supplementary figure and would show additional value of genomic data in surveillance.

(Remarks on code availability)

The code is very well written and documented.

Reviewer #3

(Remarks to the Author)

Verschuuren et al. have written an excellent manuscript describing the evolution of drug resistance in southeast Asian P. falciparum following changes in first-line therapies as a result of high prevalences of resistance to DHA-PQ. In line with the stated goal of clearly communicating trends to national malaria control program to facilitate appropriate policy changes, this manuscript presents easily interpretable maps and description of population changes, with sufficient contextualization to be useful for a wide range of audiences. In summary, I believe this manuscript will be extremely informative for national malaria control programs in Africa as decisions are made to address local emerging resistance.

As much as I enjoyed this manuscript, I have some requests and suggestions.

Major concerns:

1. What has happened with the PfCRT mutations that are associated with piperaquine resistance? Have they decreased in prevalence along with PM2/3 amplification or have they been more stable in the population? If this data is available, it should be included. If it is not available it should at least be mentioned in the manuscript and listed as a limitation. I really hope it can be included.

Minor concerns:

1. I recognize that mutations in mdr1 that are of interest in Africa (especially N86Y, perhaps Y184F and D1246Y) have not been linked to variation in lumefantrine/mefloquine susceptibilities in SEA, explaining why they have not been reported here. However, it might be clarifying for some readers interested in applying the trends reported here to emerging resistance in Africa to know prevalences of these mutations if the data are available (as supplemental data) and to add a short comment to the text explaining that these mutations do not appear to play a role in drug susceptibility in this population. Sorry for my

Africa-focused perspective, but what has been done in SEA will be a model for what will be done in Africa, and I see this manuscript as an opportunity to further advance utilization of molecular surveillance to inform decisions.

2. The authors state that Laos showed the first signs of decline in sample numbers, yet mention that Laos continued using AL and did not change first-line drugs. Laos also seems to be unique in showing evidence of decreasing prevalence in K13 mutations. Can the authors speculate on what may have caused these trends?

3. Would it make sense to define clusters as a proportion of samples (20% of samples?) rather than requiring a cluster have at least 20 samples? It seems like the low number of samples collected from Laos would make it less likely that clusters of 20 samples would be identified, even if a majority of the samples were closely related. Again, some Africa-centric interest in these data because of Laos' use of AL.

(Remarks on code availability)

Instructions are clear for installation, but it does take a bit of time. I got some errors because of pre-installed packages, which could be confusing for some individuals without much experience with R.

Version 1:

Reviewer comments:

Reviewer #1

(Remarks to the Author)

The authors have added additional analysis and discussion and thereby addressed most issues raised in the previous review phase. The manuscript reads well with important conclusions for drug resistance evolution and antimalarial treatment policy. I recommend to accept the current version of the manuscript for publication.

(Remarks on code availability)

I reviewed and tested the code in the first round of review, I quickly checked the github, but I do not believe it has changed much since then.

Reviewer #2

(Remarks to the Author)

The authors have responded well to the majority of my comments. I feel most importantly, they have greatly improved the description and analysis of the effects that ACT policy changes and COVID19 lockdowns had on Pf prevalence and PPQ resistance. This is the central value in the manuscript, and I am content it has been handled well now. Another important improvement is that the authors have now included critical methodological information and references in the Supplementary Methods; in particular, about how inference of PPQ resistance, analysis of crt mutations, and the sample genotyping were conducted.

I still feel the `grcMalaria` package is being marketed rather inaccurately. In the manuscript the authors say (Line 74): "The tool is easy to use ... making geospatial analyses accessible to a wide range of users with different backgrounds."; whereas, in response to my concerns that the input data format is non-standard and would effectively prevent anyone using the package outside of their team, they respond: "The primary objective of the R library was to support our partners who need to process the data in GRC format which we deliver to them.". Needless to say, these two statements are inconsistent. The argument about VCF not holding metadata is superficial – you don't need all information in one file.

I am also a bit confused why there is resistance to my point around including some plots of sequencing data quality, like coverage, in the supplementary material (i.e. major comment 4). I would assume / hope these are already being generated in-house, or should only take a few hours maximum to create from the VCF / BAMs. But if the authors think it's enough to just reference their filtering criterion, I won't push it further (even though I am quite sure it would make a lot of the more technically-minded readers happy to see).

Overall, the manuscript presents compelling longitudinal genomic-epidemiological data surrounding policy changes in GMS. I appreciate the work everyone involved undertook to make this large study a reality, and I look forward to future work from the team.

(Remarks on code availability)

Reviewer #3

(Remarks to the Author)

I thank the authors for their consideration of my suggestions. I found the additional data helpful. And am generally satisfied with your responses. I do have a couple of comments that I think would help with clarity and ease of interpretation.

My biggest concern is how the crt data has been incorporated. While the added data is very helpful and reassures me that your initial conclusions, that PPQ-R (defined molecularly) is decreasing is indeed valid, I can't help but think that your argument would be more convincingly communicated if the crt data was given a little more emphasis. My preference would be to include a more detailed presentation of the distribution of crt genotypes within the kel1/pla1 lineage presented in the main text; it feels a little buried in the supplemental and the x2 and p-value don't communicate the information to me as well as supplemental figure 9. At the very least, start this section (line 232) mentioning crt as a marker of piperazine resistance and presenting your argument that focusing on PM2/3 amplification is sufficient due to the high rates of coincidence, justifying why it is not incorporated into your "predicted PPQ-R" criteria.

My other more minor comments are as follows:

1. Legend of figure 1. Are you sure that marker size is proportional to sample size? I'm assuming the numbers in the marker represent the number of samples collected and the circle labels "1" in Quang Binh seems to be very similar in size to "59" Quant Tri.
2. For legends for figures 2 and 3, consider directing the reader to sup table 1 for the definitions of predicted resistance; even better would be to add them directly to the legend or text if feasible as accessing supplemental files can be a nuisance.
4. Providing the definition of predicted PPQ-R should be done a bit earlier (line 246 rather than 250 and 266).
5. The definition of the mdr1 haplotype could be presented a bit more clearly somewhere around line 316.

(Remarks on code availability)

No further comments;

Manuscript: NCOMMS-24-36123

Genetic surveillance of *Plasmodium falciparum* populations following treatment policy revisions in the Greater Mekong Subregion

Wasakul V *et al.*

Review Comment Responses

Dear Reviewers,

We would like to thank you for spending time reviewing our manuscript. We have addressed all your comments to the best of our ability, and we believe we have covered all the points raised. Your comments were at times challenging to respond to, but always useful and we feel that the manuscript is much improved as a result of the changes made, particularly the additional analyses requested. We hope the new manuscript will meet with your approval.

Best wishes

The Authors

Reviewer #1 (Remarks to the Author):

Review GENREMEKONG paper

The manuscript describes genetic surveillance of drug resistance of *Plasmodium falciparum* in the Greater Mekong Subregion, and presents promising results. It shows a remarkable decrease in plasmepsin 2/3 amplifications, from a prevalent 62% to a mere 2%. While artemisinin resistance remains high, the absence of resistance to partner drug mefloquine in Cambodia is encouraging.

Strengths:

The study includes a large number (5982) of dried blood spot samples from multiple provinces in 3 different countries from 2017-2022. The authors applied a well-standardized genotyping protocol, both in the lab and in silico, which have been made openly available. Results can have important implications for the field.

Major Concerns:

1. Conclusions on DHA-PPQ Resistance: The strong conclusions regarding the disappearance of DHA-PPQ resistance in the GMS are concerning. The authors based their predictions of PPQ resistance on plasmepsin 2/3 amplifications alone, disregarding strong evidence from the same region of emerging mutations in the *crt* gene, which are also validated markers of PPQ resistance according to the WHO. The evidence for *crt* mutations includes a study from this journal that demonstrated that mutations in *crt* emerging in Cambodia caused in vitro resistance to PPQ, also without the presence of the plasmepsin 2/3 amplification (Ross et al. Nat Comms 2018). Another clinical study published in the Lancet Infectious diseases journal, including co-authors from the current study, confirmed the association of these *crt* mutations with DHA-PPQ failure in Cambodia (van der Pluijm et al 2019, Lancet Inf Dis). While in the clinical study the *crt* mutations were seen in KEL1/PLA strains with pm2/3 amplifications, from the loss of these amplifications alone in the current study, the authors cannot conclude that the parasites are susceptible.

Without genotyping data of this region of the *crt* gene and the lack of phenotypic characterization of the strains without pm2/3 amplifications, it cannot be concluded that DHA-PPQ resistance is no longer present in the studied area. The conclusions made by the authors could lead to misinterpretations with severe consequences for clinical practice.

It would be valid to conclude that there is resistance in an area where you encounter only one of

multiple resistance markers. It is however very risky to conclude the return to susceptibility based on the absence of only one of multiple validated makers.

We would like to thank the reviewer for their insightful comment regarding *crt* mutations. The GenRe-Mekong platform genotypes most of these *crt* mutations, but we had disregarded them in the narrative since they are almost never found without *plasmepsin2/3* (*pm2/3*) amplifications. This was a shortcoming, which we have now corrected.

To address the reviewer's concerns, we conducted additional analysis on the prevalence of validated markers of PPQ resistance in the *crt* gene, to assess whether there has been an increase in prevalence of *crt* mutations without concurrent *pm2/3* amplifications. Four of the *crt* loci (T93S, H97Y, I218F, G353V) were available in the genetic report card produced by our amplicon sequencing pipeline. Of the 5,982 samples presented in the manuscript, 3,066 samples had available genotypes at these positions. We also analyzed additional *crt* markers that have been associated with piperazine response, by inspecting whole-genome sequencing data which was available for a subset of samples (n=1581), which enabled the genotyping of additional loci at F145I and M343L.

Our analyses revealed that in the prevalence of these *crt* mutations did not increase in the absence of *pm2/3* amplifications, confirming that these *crt* mutations do not circulate independently ($\chi^2 = 245.16$, $p < 0.001$). Also, we observed a clear correlation between the decline of *crt* mutation 218F and the decrease in *pm2/3* amplifications. Taken together, evidence available from *pm2/3* and *crt* mutations clearly points to a real decline in PPQ resistance, supporting the conclusions presented in the original manuscript.

This new analysis has been included in the revised manuscript and additional materials.

2. Presentation of Data: The way the data is presented could lead to misinterpretation of the data. For example, the multipanel figures showing pie charts that are proportional for the sample sizes do not clearly indicate the differences in sample sizes between the periods.

The size of pie-charts is only proportional to the sample size for the province within the time period, and it is not proportional between the different panels (i.e. different periods). Due to the big differences in sample size this should be improved.

We have corrected the presentation of data according to the reviewer's suggestions.

3. Generalization of Findings: The manuscript makes associations between treatment policy changes and changing population dynamics, while the data does not strongly support this statement. The treatment policy only changed in Vietnam during the study period, while the populations in that country did not change (based on the barcode data). Findings in one country should not be so strongly generalized for the region, especially considering the fragmented sample collections and differing results and treatment policies in the three countries in this study.

The above high-level comment is fleshed out in more detailed comments by the reviewer further below. We have responded in details to these comments, and we feel that the changes made to address each comment, when combined, address the comment above. Our conclusions are now bolstered with observations from earlier studies in Cambodia, which confirm the analyses performed in Vietnam; and we explain the epidemiological changes in Laos. Please refer to the new versions of our results and discussion.

4. Influence of External Factors: The influence of external factors, such as restrictions during the COVID-19 pandemic, on malaria incidence, sample collection, and observed patterns is not fully addressed.

In conclusion, more nuance is needed in presenting and interpreting the data throughout the manuscript.

We have included an analysis to assess the impact on *P. falciparum* prevalence and PPQ-R of (a) COVID19 lockdown measures (as reported by official sources), and (b) of the switch in frontline ACTs, using segmented regression analysis. Although both interventions impacted the decline in prevalence, the change of frontline ACT in Vietnam had a more pronounced and statistically significant impact on prevalence reduction and resistance. We have also shown, referencing another study, that the rapid drop of prevalence in Cambodia did coincide with a change in frontline drug, preceding the pandemic lockdown restrictions, which showed no significant effect on prevalence in Cambodia and Laos. These analyses have been added to the revised manuscript.

Specific major comments

- Line 122-123 - The groups are not very even this way. It might make more sense to make two groups: 2017-2019 vs. 2020-2022. Any specific trends in 2022 you would want to highlight, you could do so in the text.

We agree that the three groupings (2017–2019, 2020–2021, and 2022) do not evenly divide our dataset, but we believe that this split provides critical analytical benefits. Specifically, the period from 2020–2021 reflects a unique phase characterized by policy changes and external disruptions (e.g., pandemic-related impacts), which makes it distinct from both the earlier years and the year 2022. By separating these periods, we can more effectively capture and highlight the nuances of temporal trends and assess how these factors influenced outcomes during this transitional period. Combining 2020–2022 into a single group would obscure these unique patterns and make it more challenging to interpret the data within the context of these pivotal events.

o This way also the difference in sample size will become more clear in these figures. For example in 2022, the size of the piechart from Attapeu (59 samples) is only very slightly larger than Quang Tri with only 3 samples. In 2017-2019 the size of the piechart of Quang tri is the same as in the later years, but in 2017-2019 you had 64 samples for this province, while in 2020-2021 there were only 2 and in 2022 there were only 3 samples.

As mentioned above, we have now corrected the presentation of data according to the reviewer's suggestions.

- Line 165 – Figure 1A does not show the incidence of malaria in Vietnam; it shows the number of samples collected. From your samples you probably cannot calculate/estimate incidence and you will need to refer here only to the data reported to the WHO.

We thank the reviewer for their suggestion, which we have implemented.

Figure 1a now shows the number of Pf samples collected by the GenRe-Mekong project, as well as Pf prevalence, calculated as the proportion of confirmed Pf cases reported by WHO, divided by the total number of tested cases. This new figure shows a clear decline in malaria prevalence from 2019 for Cambodia and Laos, and from 2020 for Vietnam. The number of samples collected in Vietnam is consistent with the country's longitudinal prevalence trend, indicating that- to a first level of approximation- routine surveillance in Vietnam can be an indicator of Pf prevalence in this country.

- Line 168 – “As incidence declined over time, Pf populations became more geographically isolated.” Based on what results/outcome variable is this statement based? It is not clear on the basis of the results presented so far.

This statement is a simple observation that the spatial distribution of cases changed over time, such that by 2022 they were concentrated in within a single province per country, as shown in

Supplementary Table 2. We realize that the choice of word may not have been ideal, so we rephrased- the text now reads:

“As the number of cases declined over time, *Pf* populations became more spatially patchy. By 2022, the majority of samples were concentrated in one province in each country [...]”

- Lines 172-191 and Figure 2 and Figure 3 => PPQ resistance was predicted only on the basis of PM2/3 amplifications. The manuscript should clearly state that other markers of PPQ resistance were not investigated.

As mentioned earlier, the revised version includes a full analysis of piperazine-associated *crt* mutations.

- Lines 177-181 “Although KEL1/PLA1 began to diversify as it spread, it maintained its two characteristic 178 haplotypes during this period: a kelch13 C580Y mutant haplotype conferring artemisinin resistance, 179 and a pm23 amplification conferring resistance to piperazine. This strain also spread into southern Laos, likely as a spillover from neighboring countries, producing high levels of DHA-PPQ resistance in 181 Attapeu and Champasak Provinces, despite the ACT not being used nationally.” => **where are these results? It is not clear to me from table 1 or figure 5 that the lineages in Laos were a spillover from neighboring countries?**

Evidence for this is discussed in previous studies in Laos (Jacob *et al.*, 2021; Wasakul *et al.*, 2022), which are now cited in this section. Additional support is provided in the present manuscript, as shown in Figure 6, illustrating the spread of DHA-PPQ-resistant lineages prevalent in Vietnam that were also present in Laos during 2017–2019. In any case, we removed references to the spill over hypothesis, although it is consistent with current views of the spread of DHA-PPQ (also described by Imwong *et al.* 2017, cited in our manuscript).

- Line 183-185 “Following these changes, 184 regional DHA-PPQ predicted resistance fell steeply, to 30% (204/690) in 2020-2021, and 1% (2/278) in 2022 (Figure 2a, Supplementary Figure 3a).” => Please add statistics to support this statement, especially considering the huge difference in sample sizes: 62% (1,957/3,132) to 30% (204/690) to 1% (2/278). You can also see that the confidence interval in figure 3 for the ‘PPQ-R’ in 2021 and 2022 is quite wide.

A Cochran-Armitage trend test was performed to evaluate the trend in *pm2/3* amplification proportions across the three periods (2017–2019, 2020–2021, 2022). Accounting for differences in sample sizes, the test identified a significant decline in amplifications ($Z = -24.528$, $p < 2.2e-16$). These results have been incorporated into the text.

- Line 185-188: “To demonstrate that these changes in first-line treatment policies were the likely cause of DHA-PPQ resistance decline, we plotted the prevalence of predicted piperazine resistance (i.e. the *pm23* amplification prevalence), alongside the timeline of first-line treatment strategies for uncomplicated *Pf* in Vietnam (Figure 3).” I am not sure that this is the right conclusion to make, it is circumstantial at best.

o There are several issues here:

- When plotting the *pm2/pm3* amplifications, I would only do so for the provinces where the treatment policy has actually changed.

Thank you for the suggestion, which we have implemented. Only the provinces where the treatment policy changed are now used for the plot.

- I would change the legend of the blue line to better reflect that this is only based on PM2/3 and not all validated markers.

We have modified the text in the legend to reflect this. It should be noted that our new analysis of *crt* mutations suggests that these were of not consequence anyway.

- Why only analyze this for Vietnam? And not for the other countries?
- As far as I am aware Pyramax was already used prior to 2020 in several of these provinces in Vietnam. Dak Nong and Binh Phuoc changed to pyramax as first line treatment mid-2019, after results from TES trials with pyramax were analyzed (conducted in sentinel sites in several provinces in 2017 and 2018). In Dak Lak and Gia Lai treatments changed early 2020.

The Vietnam Ministry of Health officially announced the use of Pyramax in five endemic provinces towards the end of Q1 2020 (Vietnam Ministry of Health, 2020, referenced in the manuscript). We relied on this official information.

- There were a lot of other things going on in 2020/2021 in Vietnam that have potentially impacted malaria transmission and sample collections, including extensive strict limitations to cross-border travel between provinces due to COVID outbreaks and lock downs preventing people to leave the house for example to go into the forest or fields where they are more at risk of infection.
 - Line 188-191: “We observed that within 3 months after replacing DHA-PPQ with AS-PYR in 5 provinces, a reduction in Pf incidence and a decline in PPQ 190 resistance began, which within 18 months led to the disappearance of piperazine-resistant parasites 191 (Figure 3).”
 - o Same comments as raised before: nr of samples collected is not the same as incidence. Especially considering barriers to patients seeking access to healthcare and barriers to sample collections during restrictions as a result of COVID.
 - o AS-PYR was not implemented at the same time in all provinces.
 - o PPQ resistance based on PM2/3 alone

We believe that the revised manuscript, with additional analyses of relative impact, and of *crt* mutation prevalence, has addressed these comments.

- Line 194-195: “presumably because artemisinin-based combinations continued to be the first-line treatments in the region.” Avoid presenting speculations in the results section.

Thank you for this recommendation. This sentence has been removed.

- Line 195-199: “However, in provinces at the border of the endemic region (Savannakhet in Laos, Quang Tri and Ninh Thuan in Vietnam), most parasites were predicted to remain susceptible to both artemisinin and piperazine throughout the study period (Figure 2b), suggesting that the local parasite populations were isolated from the spreading resistant strains in the eastern GMS.” I am not sure that Ninh Thuan qualifies as “the border of the endemic region”, and that this is the reason why the parasites in this region are different. The coastal provinces have historically had different malaria populations and factors driving transmission than the Central Highland provinces, through a complex interplay of environmental and socio-economical factors. There are probably other factors at play both here and probably in other provinces as well that contribute to the genetic patterns observed rather than just being at the edges of the (current) endemic regions.

The sentence may have been over-interpreted in this case. It was not suggesting that being “at the periphery of the endemic region” (we have modified the text) is the cause of isolation of these populations, it was merely stating that this is where they are located. We were just observing that, although the DHA-PPQ-R sweep had spread pretty much across the whole territory we monitored, there were populations that remained isolated from that sweep.

- Line 218: It is interesting that C580Y remained the dominant haplotype in Vietnam, while both in Laos and Cambodia this really changed.

- Lines 227-232: “Following the implementation of new front-line treatment policies, however, a noticeable shift in population dynamics occurred. In Vietnam, the KEL1/PLA1-derived clusters KLV01 and 232 KLV03 continued to circulate but lost their pm23 amplification.”

o The statement regarding the shift in population dynamics following the implementation of new front-line treatment policies is not strongly supported by the data.

o The treatment policies only changed in Vietnam in the study period, while the population/the barcode in the Vietnam samples did not really change, but this strain ‘just’ lost its PM2/3 amplifications and became predominant. This indicates evolution of this lineage, not a shift in population dynamics.

This is a valid observation, and this may have been an unfortunate choice of terms on our part. We have now simplified the sentence, removing references to “population dynamics”

o In Cambodia, treatment policy already changed in 2017, but the population in Pursat changed in 2022 compared to 2020-2021 . There are no samples from Pursat from 2017 to 2019, in addition, the sample size in 2020-2021 is much lower than in 2022 in this province. The evidence for a population shift directly due to the change in treatment policies is weak.

Although there are gaps in our surveillance data, there is evidence from other studies that corroborates our hypotheses.

Our Figure 1 shows that Pf prevalence in Cambodia started to decline rapidly in 2018 (WHO data) which fits well with the change in treatment policy starting in 2017. What is remarkable, however, is that up that point KEL1-PLA1 was enjoying massive growth. Hamilton *et al.* (2019, LID) shows that KEL1/PLA1 was found in 75-100% of parasites in multiple parts of Cambodia in 2016-2018 (see figure below). Thus, the massive drop in prevalence in 2018-2019- well before the pandemic- was not only very rapid and geographically widespread across the country: it was also specific, in that an amazingly successful strain that had almost reached fixation was quickly wiped out, and replaced by other ART-R strains.

[REDACTED]

We think this strongly supports our hypothesis of an epidemiological change driven by treatment policy change, better than any alternative theories about movement restrictions. The discussion now contains a paragraph presenting this evidence, and explaining it more coherently.

o In Laos, the treatment policies have remained the same throughout the entire study period, so a change in treatment policies cannot be the explanation for the emerging(?) KLV04 and KLV07 lineages.

In previous work, we have accounted the emergence of KLV04 and KLV07 in Laos (Wasakul *et al.*, LID, 2022). Though Laos did not change treatment policy, the end of DHA-PPQ use in neighbouring countries weakened drug pressure sustaining the KEL1/PLA1 strain. It is certainly also likely that movement restrictions in 2020, due to the COVID-19 pandemic, reduced the number of imported parasites.

That publication also showed that KLV04 and KLV07 lineages are close descendants of artemisinin-resistant lineages found before 2010 in Cambodia, which persisted over long periods at very low frequency. Since KEL1/PLA1 was a strain imported to Laos (Imwong *et al.* 2017, Hamilton *et al.* 2019), as its frequency dropped in Cambodia it also waned in Laos. This resulted in a rise in frequency of KLV04 and KLV07, which are probably fitter in this new scenario.

This has now been clarified in the discussion.

- Line 232- 234: “By the third 233 quarter of 2021, none of the circulating clusters possessed the pm23 amplification, and the Central 234 Highlands of Vietnam were dominated by the piperazine-susceptible KLV03 population (Figure 5).” – is there any clinical or phenotypic data to support that the KLV03 population is piperazine-susceptible? The absence of pm2/3 amplifications is not sufficient for this conclusion.

The project does not collect clinical or phenotypic data. However, we have now analyzed both *pm23* amplifications and *crt* mutations, with concordant evidence that provides support for a regional decline in PPQ-R prevalence during this study period.

- Line 235-237: “Taken together, these results suggest that the observed reduction in DHA-PPQ resistance in 2020- 236 2021 can be explained by the disappearance of KEL1/PLA1 clusters, and their loss of pm23 237 amplification, occurring after changes in first-line treatment policies.”

o There is no reduction in DHA resistance => still very high levels of K13 associated mutations.

Correct. This is why we specified DHA-PPQ resistance, as opposed to artemisinin resistance.

o KEL1/PLA1 clusters have not disappeared in Vietnam, they have changed. The clusters are identified based on the barcode, which did not change, but they have lost the pm2/3 amplification.

We now have changed the sentence to “explained by the disappearance of KEL1/PLA1 clusters, or their loss of pm23 amplification”

o The data is incomplete to suggest that there is no more PPQ resistance. Genotypic data for *crt* alleles associated with PPQ resistance are missing, nor is there any phenotypic or clinical data to support this.

As detailed above, this has now been rectified.

o Clusters have changed in Cambodia and Laos, but the link to changed treatment policies is not that clear (see earlier comment).

As detailed above, this has now been addressed in the discussion.

- Line 248-249: “Our data shows that dramatic epidemiological changes occurred following changes in first-line treatment policies in the region.” – The study design is not appropriate to determine

epidemiological changes.

- The discussion should be adjusted based on the requested nuances and changes in interpretation highlighted in the comments to the results section and main concerns section above.

We appreciate the reviewer's insights. Our study's strength lies in its systematic routine surveillance approach with dense territory coverage over a long period of time, which we believe is one of the most robust methods available for capturing epidemiological trends. While we acknowledge that observational studies cannot definitively establish causality, our design is exceptionally well-suited to monitor temporal and regional changes in parasite population. We have revised the discussion section to more explicitly address the nuances associated with our findings.

Minor comments:

- Ensure the number of provinces in Vietnam is consistent throughout the manuscript.

o Where are the results from Quang Nam province?

For the period covered, we received no *Pf* samples from Quang Nam province.

The sample collection framework in Vietnam was designed and managed public health officers at IMPE-QN, which oversees malaria activities in 15 central and highland provinces where the disease is most endemic.

- Address the limitations in sample collection and the potential impact on the results.

o Sample collection was limited to specific provinces, and not all provinces sampled in all years. potentially missing variations in other regions and years.

Samples were collected in all provinces targeted by the National Malaria Control Programmes with whom we were collaborating, and whose support is the objective of the GenRe-Mekong project. Naturally, the pace of implementation was different in different countries, which meant that longitudinal coverage is uneven.

We would like to point out, however, that the decline in *P. falciparum* cases in the later part of this study period resulted in very uneven sample yield. Hence, the lack of data from some provinces reflects a lack of malaria cases, rather than a failure of the collection effort.

We have added this limitation in the discussion.

o The study relies on samples from symptomatic patients, which may -not represent the entire population, including asymptomatic carriers.

We agree that the samples used in this study do not represent the asymptomatic population. This was a design choice early in the course of the project, based on increasing genotyping failure rates at lower parasite densities. Our amplicon sequencing platform works well at parasitaemia levels that are detectable by RDT, which are on average higher than those in asymptomatic infections in the eastern GMS. As a result, it is most suited to surveillance of clinical cases, which makes the design of a sampling framework more practical for control programmes.

Given the technical and operational complexity of characterizing asymptomatic infections at this level of genetic detail, vis-à-vis the value that our data provides, we think the trade-off is reasonable. Through its collaborations, GenRe-Mekong was able to capture and genotype a substantial proportion of *Pf* cases reported to WHO.

We have added this limitation in the discussion.

- Line 106-109 – please provide references or details for the different PCRs used.

References have now been added.

- Line 127-129 – Clarify how allele proportions were determined and how multiple clone infections were taken into account.

We have added a sentence in the method to cover this.

- Line 135-138: Include details on the Louvain algorithm in the methods section.

A reference has been added.

o How are missing positions imputed? Is any other data used for the imputation, for example, is imputation done at each province level?

Details on imputation have been added to the Methods, in section “Analysis of highly related parasites”.

o How does your algorithm with at least 20 members impact the interpretation of the data from 2022, where you do not have 20 samples from many provinces. You mention somewhere in the results that the provinces are becoming isolated, but this is difficult to conclude as a large part of your samples in 2020-2021 and 2022 do not belong to a cluster. Considering the rules you applied for the clustering, with such low numbers of samples, can the absence of clustering be interpreted as increased isolation?

We agree that our initial clustering threshold of 20 members might have led to the exclusion of smaller clusters. To address this, we have revised our method to form clusters with a minimum of 10 members, while retaining the same similarity criteria. This adjustment allows us to capture smaller clusters; the interpretation of the result did not change with this adjustment. We are reluctant to make the threshold lower than 10, as there is a risk of over-clustering due to sampling artefacts.

Regarding the observations about the distribution of clusters, perhaps the word “isolated” is misleading. We were merely observing that clusters in recent years tend to remain geographically confined, in contrast to the more widely dispersed clusters observed during 2017–2019. We have changed this in the text as follows: “From 2020, clusters remained confined within single countries, often restricted to single provinces [...]”

- Lines 238-244: these results are very separate from the main aims of the manuscript and it is unclear why they are added here. If it to present and demonstrate the software-tool, then the development of this software tool needs to be better described in the methods. What does this tool do and why is it different from the tool used for example in the Jacob et al. paper.

We have removed that section.

The tool presented there was the R package that the reviewer describes in the section below. This tool did not exist when Jacob *et al.* was published. We had used Tableau, a commercial software application, to visualize prevalence, but without the analytical capabilities provided by the grcMalaria R library.

Reviewer #1 (Remarks on code availability):

An R package is available on github and a userguide and tutorial is available on <https://genremekong.org/tools/grcmalaria-r-package-user-guide>.

The package is very useful to the community and allow replication of the analysis. In addition, the software package can be used to analyse your own Genetic Report Cards.

I tested the package by downloading and running parts of the example notebook

We would like to thank the reviewer for the positive feedback and for testing the library.

Reviewer #2 (Remarks to the Author):

Verschuuren et al. summarise P.f. drug resistance genotypes from ca. 6000 samples collected across the Greater Mekong Subregion (GMS) from 2017 – 2022. The genotyping data is generated using a previously published protocol & bioinformatic pipeline, and visualisations are made using a spatial mapping software (grcMalaria) the authors developed in R and have now made public.

The manuscript adds to a growing number of studies where genomic surveillance of P.f. is being used to aid and analyse policy decisions, and deploys one of now at least three existing multiplexed amplicon sequencing assays for Illumina. The paper continues from previous work where the authors sequenced ca. 10000 samples across the GMS using the same approach (<https://elifesciences.org/articles/62997#content>). Compared to related literature the work benefits from scale (e.g. more samples and geographical coverage) and a greater commitment to making data interpretable for policy makers through conversion to phenotype and visual aids. In my opinion, SpotMalaria/GenRe-Mekong currently represents the most complete effort to systematically realise the translational value of P.f. genomic data in policy.

That being said, from a methods point of view, I found no improvements in this manuscript over previous work. In addition, I believe the bioinformatics pipelines may now be outdated, and I was unable to find key validation experiments or sequencing quality control data (in this or previous manuscripts). Moreover, my opinion is the grcMalaria software is unlikely to be used by the wider community due to its complex input requirements.

The main value of the paper is epidemiological: in the presentation of longitudinal resistance trends in response to switching first-line treatments across the GMS. The authors make some interesting / important observations, but very similar observations have been made previously by an overlapping team (<https://doi.org/10.1128/aac.01121-21>), and I feel in this manuscript they need to be more thoroughly analysed and contextualised.

Thank you for the evaluation. We agree that the manuscript is primarily intended to provide an epidemiological update. For this to be possible, the genotyping platform had to remain the same. As the reviewer points out, there are now alternatives being used by other groups. Without going into the relative merits or defect of each of these platforms, we would like to stress that their overlap with the GenRe-Mekong platform is limited, and this would have seriously impacted our ability to perform genetic epidemiology analyses, and to provide our partners with comparisons between different time periods.

Major Points

1. The main finding from the manuscript is that “...dramatic epidemiological changes occurred following changes in first-line treatment policies across the region.” (Lines 247 – 248). In my view there were two major potentially relevant policy changes: switching ACTs and COVID19 restrictions; and two dramatic observations: declining prevalence and PPQ resistance. While I appreciate that establishing causal links is challenging, the data supports much more investigation into these relationships than has been presented. For example, Fig 3 presents data only for Vietnam; Cambodia and Laos underwent different policy changes (earlier switch to AS-MQ; or retaining AL) and also investigating prevalence/PPQ-R trends

here (potentially incorporating your prior data for a longer timeline) could strengthen evidence for first-line treatment being a causative factor (especially given the increased geographical isolation you observe might imply limited parasite migration). Relatedly, note that Lines 62 – 63 vs. Lines 181 – 182 give a contradictory timeline for the policy change in Cambodia. I also suggest that annotating Fig 3 with relevant COVID19 restrictions would help readers more intelligently interpret your main findings. At present, I believe more analysis can be done to support the main claim, and to try and disentangle COVID19/ACT policy effects (which the authors only mention briefly Lines 268 – 270).

We appreciate the reviewer's insight. We have listened to your suggestions and have performed additional analysis to assess impact of COVID19 lockdown and switching in ACTs on *P. falciparum* prevalence and PPQ-R using segmented regression; this analysis could only be conducted in Vietnam, where our data supported trend analyses before and after the two interventions. While both interventions impacted the decline in prevalence, the switching ACT in Vietnam had a more pronounced and statistically significant impact on prevalence reduction and resistance. The effect of ACT switching could not be assessed in Cambodia and Laos, however, lockdown restrictions showed no significant effect on prevalence in both of these countries. A new Supplementary Figure 6 had been created to show changes in *Pf* prevalence and pm2/3 amplification frequency with timeline for major policy changes in the three countries.

2. A potentially high-value output is the grcMalaria package which was used to generate the spatial maps of resistance in the paper. I reviewed the github repo and commend the authors on the high code quality and extensive documentation. However, I highlight that the utility of grcMalaria to the scientific community is severely limited by the fact that it requires a very complicated-looking and non-standard input data format (i.e. the Genetic Report Card) that is produced by their upstream pipelines. Despite providing a description of the columns, practically speaking it would be very laborious / error-prone to convert from standard genotype output formats (e.g. VCF) into this excel spreadsheet. To rectify this, the authors need to support standard / simple input file types (VCF, a CSV with sample ID and collection location); e.g. by providing an R function to convert from standard input file types to GRC format. Furthermore, could you comment on how grcMalaria would work with amplicon sequencing data from other assays? As it stands, I believe grcMalaria will see little use by the wider community.

We respect the reviewer's opinion. The primary objective of the R library was to support our partners who need to process the data in GRC format which we deliver to them. The GRC format has been in use by our partners for quite a while, long before the library, and it was designed to be usable by public health officers, who can use Excel spreadsheets and generally understand the main columns (time and place, drug resistance predictions), while not being well versed with genetic data. A VCF file would please bioinformaticians, but would be far more difficult to use for public health staff, who are not trained in genomics; furthermore, VCF would be rather unsuitable for encoding sample metadata, forcing us to distribute multiple files. The Excel format, compared to CSV, gives us an opportunity to use formatting styles, e.g. to encode drug resistance status using a "traffic light" scheme that improves interpretation. Again, all this was intended to make the data more usable by non-experts of genetics.

We are currently developing the "next generation" of GRC format, which is "self-descriptive" - in other words, one will not need to use a predefined set of columns, but the GRC file itself will contain a description of the columns. This will make the GRC format more flexible to be adopted by other genotyping platforms; we will leverage this for our *P. vivax* platform, currently under development. We hope that this will encourage the broader research community will consider using our software, but it should be stressed that our primary objective is to support the surveillance partners.

3. In the Methods, the authors cite their previous publication (<https://elifesciences.org/articles/62997#content>) and state they only summarise salient points in this manuscript. However, several salient points are missing. Critically, I can't determine clearly how PPQ resistance was called, despite that being central to the manuscript's conclusions. The methods say that qPCR was "also used" to detect amplification, but no specific qPCR assay is cited. The primary detection was presumably by amplicon sequencing, but in the technical notes of the previous paper for 'Piperaquine' they mention targets against a breakpoint fusion amplicon in pm23 (which would be impressive if it worked, and would certainly require some validation) as well as novel targets in pfCRT and pfExo. How was data from these various targets and assays combined to make a final prediction of PPQ resistance? This needs to be clearly described in the main methods. Also, a basic description of the bioinformatic pipeline used to process all the sequencing data should be included.

Our prediction of PPQ-R was based on the presence of the *pm23* amplification, as detailed in the phenotype prediction rules included in the supplementary materials of Jacob *et al.* *Crt* mutations were not considered in the prediction rules, based on the observation that they very rarely occur in the field in the absence of the Pm23 amplification. However, we have now added to the present manuscript an analysis of relevant *crt* mutations to confirm our predictions.

Breakpoint sequence identification of *pm23* was detailed and validated in Amato R *et al.* (2017). We have added citations and details of how PPQ-R genotypes were determined from two genotyping methods. The generated genotypes were used to predict resistance, based on a published set of rules (Supplementary Table 1).

4. The authors processed ca. 6000 samples of which ca. 900 were excluded due to high missingness. Aside from these numbers, the authors provide no data at all on sequencing performance or quality control. For example, it is standard practice to provide data on depth of sequencing across samples and amplicons (see Fig. 1 <https://www.nature.com/articles/s41467-024-46535-x>, Fig.

1 <https://www.nature.com/articles/s41467-023-39417-1>, or Fig.

2/3 <https://www.nature.com/articles/s41467-024-45688-z> as recent examples from this journal).

Currently, readers have no way of evaluating the quality of the sequencing data that was generated, but rather have to take the final genotype / phenotype predictions on trust. Sequencing QC data should be included in the supplementary materials.

The SpotMalaria Technical Notes, referenced in the Methods and downloadable at <http://ngs.sanger.ac.uk/production/malaria/Resource/29> provide details of the quality threshold used for filtering sequencing reads, while the genotyping method (now added to the Supplementary Methods, see #5 below) described the read coverage requirements for calling genotypes. Furthermore, as stated in the same section, in our dataset release we provide for each sample a *genetic barcode missingness* estimate, which is a measure of sequence quality (missingness increases as genotyping quality decreases, typically due to low DNA concentrations in the sample).

5. From the technical notes of the previous publication I believe that the genotyping is performed with bcftools. For Illumina amplicon sequencing data there are now many tools I understand to be more suitable / accurate for the variable ploidy seen in P.f. infections (DADA2, SeekDeep, HaplotypR). Is there a particular reason that bcftools is being used over these other methods, given that it (incorrectly, in this case) assumes a diploid genome with alleles at equal frequencies? I could not find any validation of its performance on polyclonal infections in the previous manuscript, or an indication of rates of polyclonal infection in this dataset. My impression is that it will incorrectly call samples as homozygous if the minor allele within the sample is ca. <20%. I currently can't assess what impact this might have on overall results.

Thank you, there are some important points here, and the reviewer is justified in raising them. We reviewed the SpotMalaria Technical Notes which we were referencing and agree that the genotyping procedures are not well described, which can lead to erroneous interpretations- for example, we do not use genotypes from bcftools, and never have done, for the exact reason highlighted by the reviewer: Pf is not a diploid. Instead, we use bcftool to extract the read counts for each allele, and call genotypes from these counts.

We can certainly consider migrating to new tools in the future, and we are currently spearheading this transition by using a DADA2-based pipeline for our new Pv pipeline.

We have now added one page of Supplementary Methods, to describe the genotyping; we would also like to inform the reviewer that the genotyping pipeline is available openly on GitHub.

6. In several places throughout the manuscript (e.g. Line 164 – 165 and Fig 1; Fig 3) the number of samples collected (or passing QC) seems to be used as a proxy for P.f. clinical incidence / prevalence. While I understand you are sequencing a considerable percentage of all cases and there is a correlation here, this approach introduces unnecessary noise / confounding. Please use standard measures (prevalence / clinical incidence) when making statements / display items about transmission intensity. We have added malaria prevalence, calculated as the proportion of confirmed Pf cases reported to WHO over total number of tested cases, in the manuscript as well as on Figure 1.

Minor comments

Figure 1a: y-axis label is missing from. Moreover, adding the longitudinal prevalence trend to this figure would improve it, as it is a very important piece of contextual information that explains the reduce sample collection numbers.

Thank you for the suggestion. We have fixed the y-axis label and added longitudinal prevalence trend, calculated as the proportion of confirmed Pf cases reported to WHO over total number of tested cases.

Figure 2: PPQ-R should also be shown. Can be sympatric circulation of ART and PPQ in same site, as is one cannot infer pattern of PPQ-R resistance. I don't see PPQ-R spatial map in supplementary either but this is the key genetic change.

Maps of PPQ-R have been added to the supplementary material.

Supp. Fig 3a: I think this is panel communicates some of your most interesting results very nicely (esp. if you include DHA-PPQ as well). Consider incorporating in main text.

This figure is now Figure 3 in the main text.

Lines 238 – 244: This section (Use of grcMalaria for non-GMS data) should really not be in the main results text in my opinion. The fact that grcMalaria can also make plots for other geographies is a detail of software capabilities, not a result. It could be added to the github repository / documentation, or to the 'Data analysis and visualisation' section of methods.

We acknowledge that including this result in the main manuscript may have been out of place. This text and its corresponding figure have been removed from the main text and added to the 'Data analysis and visualisation' section as suggested.

Lines 296 – 298: The claim is made that the results generated by the authors approach (e.g. local sample collection and DNA extraction, shipping internationally for sequencing and data analysis) was more efficient, cheaper, and reduced turn-around time relative to TES. While in principle these things can be true, in practice they can be very hard to realise. Can you support these claims with evidence, e.g. do

you have numerical data on costs and turn-around times for the samples included in the study? Including data to support these claims in a supplementary figure would strengthen the paper; without it you can only claim this in principle.

We acknowledge that the practical implementation of genetic surveillance can vary. However, the potential advantages of molecular surveillance over TES in terms of cost, efficiency, and turnaround time have been outlined by others (e.g. Nsanzabana C *et al.*, 2018). While our study does not provide direct cost comparison, we emphasize the role of genomic data in complementing TES for resistance monitoring and have cited this reference to support our claim.

Line 143, Data availability: Following links here I found one GRC (v1.2) with samples going only to 2021. Is the data as described from Lines 143 – 152 currently publicly available (at <https://github.com/GenRe-Mekong/Data>)? If not, when will it be made public?

The full dataset will be made publicly accessible online upon publication.

Line 34/58: Here you state DHA-PPQ was previously dominant in eastern GMS, but my understanding from previous work was that it is primarily found in Cambodia or lower GMS? E.g. Figure 3 of <https://elifesciences.org/articles/62997#content> : the two coastal sites you have in Vietnam, and several sites in Laos and at the same longitude as DHA-PPQ-R populations in Cambodia are sensitive. Very minor point but good to keep a consistent / accurate story around location.

In the period leading to 2018, the KEL1/PLA1 strain, and its closely related derivatives, had extended from Cambodia into northeastern Thailand, the southernmost provinces of Laos, and most of Vietnam (see Hamilton WL *et al.* 2019 Lancet Infect Dis doi: 10.1016/S1473-3099(19)30392-5). Altogether, that constitutes a very large portion of the endemic regions in the eastern GMS, and thus it seems appropriate to use the word “dominant”. We do acknowledge, of course, that some areas of Vietnam and Laos remained free of these parasites.

COI data: I believe COI is being estimated by the bioinformatic pipeline using THEREALMcCOIL. It would be interesting to many in the community to see if there was a decline in fraction polyclonal or mean COI following prevalence decline around GMS. Could be a supplementary figure and would show additional value of genomic data in surveillance.

Thank you for your suggestion. We acknowledge the importance of COI in malaria surveillance and appreciate the reviewer’s perspective, but polyclonal infections become increasingly rare as transmission declines.

The COI estimates from our dataset show that the vast majority of *Pf* infections in the region covered were monoclonal (COI = 1), with very few exceptions. As a result, we could not determine any meaningful trends in COI following the prevalence decline in the eastern GMS.

We have opted not to include a supplementary figure, as it would not provide additional insight.

Reviewer #2 (Remarks on code availability):

The code is very well written and documented.

Thank you for your appreciation.

Reviewer #3 (Remarks to the Author):

Verschuuren et al. have written an excellent manuscript describing the evolution of drug resistance in southeast Asian *P. falciparum* following changes in first-line therapies as a result of high prevalences of

resistance to DHA-PQ. In line with the stated goal of clearly communicating trends to national malaria control program to facilitate appropriate policy changes, this manuscript presents easily interpretable maps and description of population changes, with sufficient contextualization to be useful for a wide range of audiences. In summary, I believe this manuscript will be extremely informative for national malaria control programs in Africa as decisions are made to address local emerging resistance.

As much as I enjoyed this manuscript, I have some requests and suggestions.

Thank you for your thoughtful and positive feedback. We are grateful for your appreciation of our work and its relevance to malaria control programmes. We are happy to address your suggestions to further improve the manuscript.

Major concerns:

1. What has happened with the PfCRT mutations that are associated with piperazine resistance? Have they decreased in prevalence along with PM2/3 amplification or have they been more stable in the population? If this data is available, it should be included. If it is not available it should at least be mentioned in the manuscript and listed as a limitation. I really hope it can be included.

We would like to thank the reviewer for their insightful comment regarding *crt* mutations. The GenRe-Mekong platform genotypes most of these *crt* mutations, but we had disregarded them in the narrative since they are almost never found without plasmepsin 2/3 amplifications. This was a shortcoming, which we have now corrected.

To address the reviewer's concerns, we conducted additional analysis on the prevalence of validated markers of PPQ resistance in the *crt* gene, to assess whether there has been an increase in prevalence of *crt* mutations without concurrent plasmepsin 2/3 amplifications. Four of the *crt* loci (T93S, H97Y, I218F, G353V) were available in the genetic report card produced by our amplicon sequencing pipeline. Of the 5,982 samples presented in the manuscript, 3,066 samples had available genotypes at these positions. We also analyzed additional *crt* markers that have been associated with piperazine response, by inspecting whole-genome sequencing data which was available for a subset of samples (n=1581), which enabled the genotyping of additional loci at F145I and M343L.

Our analyses revealed that in the prevalence of these *crt* mutations did not increase in the absence of pm2/3 amplifications, confirming that these *crt* mutations do not circulate independently ($\chi^2 = 245.16$, $p < 0.001$). Also, we observed a clear correlation between the decline of *crt* mutation 218F and the decrease in pm2/3 amplifications. Taken together, evidence available from pm2/3 and *crt* mutations clearly points to a real decline in PPQ resistance, supporting the conclusions presented in the original manuscript.

This new analysis has been included in the revised manuscript and additional materials.

Minor concerns:

1. I recognize that mutations in *mdr1* that are of interest in Africa (especially N86Y, perhaps Y184F and D1246Y) have not been linked to variation in lumefantrine/mefloquine susceptibilities in SEA, explaining why they have not been reported here. However, it might be clarifying for some readers interested in applying the trends reported here to emerging resistance in Africa to know prevalences of these mutations if the data are available (as supplemental data) and to add a short comment to the text explaining that these mutations do not appear to play a role in drug susceptibility in this population. Sorry for my Africa-focused perspective, but what has been done in SEA will be a model for what will be done in Africa, and I see this manuscript as an opportunity to further advance utilization of molecular surveillance to inform decisions.

Thank you for the reviewer's suggestion. While *pfmdr1* mutations have not been linked to lumefantrine or mefloquine resistance in Southeast Asia, we appreciate that their prevalence may be informative for researchers working in Africa. Our data show that the predominant *pfmdr1* haplotype (NFD) is strongly associated with a number of ART-R *kelch13* variants ($\beta = 3.44$, $p < 0.0001$), whereas the NYD (wild-type) and YYD haplotypes are associated with wild-type *kelch13* ($\beta = -5.14/-6.81$, $p < 0.0001$). We do not fully understand the biological and epidemiological implications of these associations, but we believe that documenting this pattern could prove useful for others. Accordingly, we have added to the Supplementary Data the prevalence maps for these mutations.

2. The authors state that Laos showed the first signs of decline in sample numbers, yet mention that Laos continued using AL and did not change first-line drugs. Laos also seems to be unique in showing evidence of decreasing prevalence in K13 mutations. Can the authors speculate on what may have caused these trends?

We think that the decline in Pf cases in Laos is attributed to a combination of intensified efforts by the Lao NMCP and the broader decline in cases observed in neighbouring countries, Cambodia and Thailand. In Laos, significant progress in malaria transmission reduction has been achieved through targeted interventions such as improved case management, vector control, and community engagement. Additionally, the use of stratified control measures based on catchment areas also played a key role in this decline. In areas nearing elimination, active surveillance strategies, including case and foci investigations, have been implemented to rapidly detect and interrupt transmission (Rotejanaprasert *et al.*, 2024).

The Upper and Lower Zones of southern Laos have distinct genetic epidemiology of *P. falciparum*: *kelch13* wild-type populations are predominant in the Upper Zone, while ART-R parasites are more common in the Lower Zone, which borders with endemic areas of Cambodia and Thailand. As a

result, it seems most likely that KEL1/PLA1 parasites, mostly found in the Lower Zone, were mostly imported from neighbouring countries (Amato et al., 2018; Hamilton et al., 2019). As KEL1/PLA1 prevalence declined in neighboring regions, its presence in southern Laos also diminished, leading to an overall reduction in *kelch13* mutant parasites.

We have modified the text in the main paper, to make this scenario clearer.

3. Would it make sense to define clusters as a proportion of samples (20% of samples?) rather than requiring a cluster have at least 20 samples? It seems like the low number of samples collected from Laos would make it less likely that clusters of 20 samples would be identified, even if a majority of the samples were closely related. Again, some Africa-centric interest in these data because of Laos' use of AL.

Defining clusters as a proportion of samples, rather than using a fixed threshold, would not be appropriate in our case. Since sample sizes vary across regions, a proportion-based threshold could lead to inconsistencies, where small sample sets produce artificially small clusters, and larger datasets form disproportionately large ones. A fixed threshold ensures comparability across regions and avoids over-clustering due to sampling artefacts. We acknowledge that our initial threshold of 20 members may have overlooked smaller clusters. To address this, we have revised our method to define clusters with a minimum of 10 members while maintaining the same similarity criteria. This adjustment allows us to better capture smaller clusters without compromising the robustness of our clustering approach. Importantly, this change did not alter the overall interpretation of the results.

Reviewer #3 (Remarks on code availability):

Instructions are clear for installation, but it does take a bit of time. I got some errors because of pre-installed packages, which could be confusing for some individuals without much experience with R.

We appreciate your feedback. We have recognized that installation can be time-consuming and that pre-installed package conflicts may pose challenges for users with limited R experience. Because of this, we have developed an interactive web tool built on the grcMalaria R package, allowing users to explore the published GRC data without requiring R proficiency (latest data will be updated upon publication): <https://genremekong.org/tools/grc-mapper>

Manuscript: NCOMMS-24-36123

Genetic surveillance of *Plasmodium falciparum* populations following treatment policy revisions in the Greater Mekong Subregion

Wasakul V *et al.*

Review Comment Responses

Reviewer #1 (Remarks to the Author):

The authors have added additional analysis and discussion and thereby addressed most issues raised in the previous review phase. The manuscript reads well with important conclusions for drug resistance evolution and antimalarial treatment policy. I recommend to accept the current version of the manuscript for publication.

Thank you for your positive feedback and for recognizing the value of our study. We appreciate the constructive input throughout the review process, which has helped strengthen the manuscript.

Reviewer #1 (Remarks on code availability):

I reviewed and tested the code in the first round of review, I quickly checked the github, but I do not believe it has changed much since then.

Reviewer #2 (Remarks to the Author):

The authors have responded well to the majority of my comments. I feel most importantly, they have greatly improved the description and analysis of the effects that ACT policy changes and COVID19 lockdowns had on Pf prevalence and PPQ resistance. This is the central value in the manuscript, and I am content it has been handled well now. Another important improvement is that the authors have now included critical methodological information and references in the Supplementary Methods; in particular, about how inference of PPQ resistance, analysis of crt mutations, and the sample genotyping were conducted.

We appreciate your recognition of the improvements in the work and glad that the expanded Supplementary Methods now provide the necessary methodological clarity. Your suggestions and insights have been invaluable in strengthening the manuscript.

I still feel the grcMalaria package is being marketed rather inaccurately. In the manuscript the authors say (Line 74): “The tool is easy to use ... making geospatial analyses accessible to a wide range of users with different backgrounds.”; whereas, in response to my concerns that the input data format is non-standard and would effectively prevent anyone using the package outside of their team, they respond: “The primary objective of the R library was to support our partners who need to process the data in GRC format which we deliver to them.”. Needless to say, these two

statements are inconsistent. The argument about VCF not holding metadata is superficial – you don't need all information in one file.

Thank you for pointing out this inconsistency. We've revised the manuscript to more accurately reflect the intended audience by changing the text to: "The tool is easy to use... making geospatial analyses accessible to users with non-technical backgrounds." Our primary goal was to support partners working with GRC-formatted data, but we acknowledge that the current input requirements may limit broader use. We appreciate the feedback and see this as an opportunity for future development to improve compatibility with more widely used data formats.

I am also a bit confused why there is resistance to my point around including some plots of sequencing data quality, like coverage, in the supplementary material (i.e. major comment 4). I would assume / hope these are already being generated in-house, or should only take a few hours maximum to create from the VCF / BAMs. But if the authors think it's enough to just reference their filtering criterion, I won't push it further (even though I am quite sure it would make a lot of the more technically-minded readers happy to see).

Thank you again for pointing this out. We agree that including data quality metrics would benefit technically inclined readers. As recommended, we have now added a sequencing coverage plot to the supplementary materials. The plot demonstrates robust coverage for all drug-related SNPs, as well as for nearly all barcode SNP positions—with only one exception where coverage is notably lower. However, even in this case, the median read depth of 30 remains sufficient for reliable analysis.

Overall, the manuscript presents compelling longitudinal genomic-epidemiological data surrounding policy changes in GMS. I appreciate the work everyone involved undertook to make this large study a reality, and I look forward to future work from the team.

Thank you so much!

Reviewer #3 (Remarks to the Author):

I thank the authors for their consideration of my suggestions. I found the additional data helpful. And am generally satisfied with your responses. I do have a couple of comments that I think would help with clarity and ease of interpretation.

Thank you for your thoughtful review and for acknowledging our revisions. We sincerely appreciate your time and constructive feedback.

My biggest concern is how the crt data has been incorporated. While the added data is very helpful and reassures me that your initial conclusions, that PPQ-R (defined molecularly) is decreasing is indeed valid, I can't help but think that your argument would be more convincingly communicated if the crt data was given a little more emphasis. My preference would be to include a more detailed presentation of the distribution of crt genotypes within the kel1/pla1 lineage presented in the main text; it feels a little buried in the supplemental and the x2 and p-value don't communicate the information to me as well as supplemental figure 9. At the very least, start this section (line 232) mentioning crt as a marker of piperaquine resistance and presenting your argument that focusing

on PM2/3 amplification is sufficient due to the high rates of coincidence, justifying why it is not incorporated into your "predicted PPQ-R" criteria.

Thank you for your comment. We've addressed your main concern by updating the text to highlight *crt* earlier in the relevant section and clarify its role as a marker of piperazine resistance. Specifically, we now state that "Although several *crt* mutations are also associated with PPQ-R, *pm23* amplification alone is sufficient for defining resistance due to its strong association with treatment failure and its frequent co-occurrence with PPQ-R *crt* mutations."

Regarding the placement of the *crt* data, we agree it is a valuable part of the resistance narrative. However, to maintain clarity and focus in the main text, we chose to keep the more detailed breakdown in the supplementary material, where it can be fully explored without disrupting the flow of the main findings. We believe this balance ensures accessibility for a broad readership while still providing the depth needed for technically focused readers.

My other more minor comments are as follows:

1. Legend of figure 1. Are you sure that marker size is proportional to sample size? I'm assuming the numbers in the marker represent the number of samples collected and the circle labels "1" in Quang Binh seems to be very similar in size to "59" Quant Tri.

To ensure label legibility, we applied a minimum marker size for data points representing fewer than 100 samples. We have removed the word "marker size is proportional to sample size" and added a marker size key in Figure 1 to clarify the relationship between sample size and marker size.

2. For legends for figures 2 and 3, consider directing the reader to sup table 1 for the definitions of predicted resistance; even better would be to add them directly to the legend or text if feasible as accessing supplemental files can be a nuisance.

We have added definitions directly to the legend.

4. Providing the definition of predicted PPQ-R should be done a bit earlier (line 246 rather than 250 and 266).

In response to the major comment, we have now provided definition of predicted PPQ-R at the beginning of the section (line 237).

5. The definition of the *mdr1* haplotype could be presented a bit more clearly somewhere around line 316.

Definitions of *mdr1* haplotypes are now included. The passage now states: "We observed that the presence of these ART-R *kelch13* alleles was strongly associated with the NFD *mdr1* haplotype (characterized by N86, Y184F, and D1246 mutations; $\beta = 3.44$, $p < 0.001$). In contrast, wild-type *kelch13* parasites predominantly carried either the NYD (N86, Y184, D1246; wild-type) or YYD (N86Y, Y184, D1246Y) *mdr1* haplotypes ($\beta = -5.14/-6.81$, $p < 0.001$) (Supplementary Figure 14)."

Reviewer #3 (Remarks on code availability):

No further comments;